# Forecast skill score assessment of a relocatable ocean prediction system, using a simplified objective analysis method

Reiner Onken[1]

[1]Helmholtz-Zentrum Geesthacht (HZG), Max-Planck-Straße 1, 21502 Geesthacht, Germany

*Correspondence to:* Reiner.Onken@hzg.de

**Abstract.** A Relocatable Ocean Prediction System (ROPS) was employed to an observational data set which was collected in June 2014 in the waters to the west of Sardinia (Western Mediterranean) in the mainframe of the REP14-MED experiment. The observational data, comprising more than 6000 temperature and salinity profiles from a fleet of underwater gliders and shipborne probes, were assimilated in the Regional Ocean Modeling System (ROMS) which is the heart of ROPS, and verified against independent observations from ScanFish tows by means of the forecast skill score as defined by Murphy (1993). A simplified objective analysis (OA) method was utilised for assimilation, taking account of only those profiles which were located within a predetermined time window $W$. As a result of a sensitivity study, the highest skill score was obtained for a correlation length scale $C = 12.5$ km, $W = 24$ hours, and $r = 1$, where $r$ is the ratio between the error of the observations and the background error, both for temperature and salinity. Additional ROPS runs showed that (i) the skill score of assimilation runs was mostly higher than the score of a control run without assimilation, (i) the skill score increased with increasing forecast range, and (iii) the skill score for temperature was higher than the score for salinity in the majority of cases. Further on, it is demonstrated that the vast number of observations can be managed by the applied OA method without data reduction, enabling timely operational forecasts even on a commercially available Personal Computer or a laptop.

## 1   Introduction

A Relocatable Ocean Prediction System (ROPS) is presented which enables rapid nowcasts and forecasts of ocean environmental parameters in limited regions. In this study, ROPS was implemented for the waters west of Sardinia (Western Mediterranean Sea) within the framework of the REP14-MED experiment (Onken et al. (2014, 2017a)).

The major components of ocean operational systems are observations and ocean circulation models coupled with data assimilation systems, to combine the observations with dynamics and issue nowcasts and forecasts which are delivered to the customers. While systems on global scale are utilised to provide estimates on large-scale circulation patterns and associated features, regional operational systems are focusing more on societally relevant oceanographic information for e.g. search and rescue operations, pollutant dispersal, fishery management (Edwards et al., 2015), and military applications. Meanwhile, quite a number of real-time ocean operational systems are available, spanning the scales of ocean horizontal circulation patterns from global to coastal (Dombrowsky (2011), Zhu (2011)).

ROPS was developed for military use in the context of Rapid Environmental Assessment but it can easily be adapted to any non-military demands. For maritime forces, there is a special need for relocatable operational systems which can be moved in any (potential) conflict area in the World Ocean on short notice. As in the majority of cases conflict areas are regionally limited, the domains of the operational systems must be tailored to the corresponding regions which means that their horizontal extent should be on the order of 100 km. Consequently, the domains share always a wet (open) boundary with the open ocean. Relocatable operational systems based on the Harvard Ocean Prediction System (HOPS, Robinson (1999)) may be considered as the pioneering work in this subject. They became available since the late 90s of the previous century, and have been applied in various regional studies up to present (De Dominics, 2014). Another line of research based on the Naval Coastal Ocean Model (NCOM, Martin (2000)) can be traced back to the first decade of the present century (Rowley and Mask, 2014), and recently, Trotta et al. (2016) presented and compared the performance of a relocatable prediction system, using structured and unstructured grids. The common properties and minimum requirements of any such system are as follows:

- A tool for setup of the model domain, including the specification of the numerical grid, the bathymetry, and the coastline,
- Interfaces for definition of initial conditions, lateral and surface boundary conditions,
- A numerical model,
- An interface for the provision of observational data,
- A data assimilation module,
- Software for post-processing of the model output.

An additional demand for relocatable operational systems is to provide accurate nowcasts and forecasts of the ocean environment in a timely manner, i.e. in near-real time. However, the requirements of accuracy and timeliness are inconsistent with one another: accuracy claims for the application of up-to-date assimilation schemes which presently are ensemble or variational methods. As the implementation of these schemes is rather complex and they are computationally expensive (Zaron, 2011), timely delivery can only be realised on powerful computers which are frequently not available. As a compromise, sequential data assimilation based on objective analysis (OA, Bretherton et al. (1976), Thomson and Emery (2014)) is used in ROPS. OA is on the one hand not as accurate as the Ensemble Kalman Filter (Evensen, 2006) or 4D-Var (Moore et al. (2011a, b)), but the computational costs are much less and the implementation is straightforward.

Meanwhile, ROPS has been implemented for various regions in the World Ocean, and it is running automatically without any major interruptions since early 2015. The concept (Fig. 1)for all realisations is identical: every day, ROPS is initialised from a restart file of the previous day's run, and it provides a 3-days forecasts relative to present. For each run, data sets for the definition of initial and boundary conditions plus observational data for assimilation are downloaded from the internet, in which the initial conditions are only required for re-initialisation of ROPS in case it died the day before.

For this article, ROPS is slightly modified because it is running in hindcast mode for a period of time in June 2014. All data for model initialisation, boundary conditions, and a huge set of observational data for assimilation are available on the local computer system and a download from the internet is not required. The objectives are to demonstrate that

- Good forecasts can be obtained from a prediction system using OA for assimilation (for the definition of "goodness", see Murphy (1993)).
- A vast number of observational data can be managed by OA without data reduction by averaging, sub-sampling, or creating "super observations" (Lorenc, 1981; Moore et al., 2011b; Oke et al., 2015).
- ROPS is able to provide timely operational forecasts even on a commercially available Personal Computer or a laptop.

The area of the ROPS model domain (Fig. 2) is characterised by a 20–50 km wide continental shelf. The shelf ends at water depths between 150 and 200 m, followed by the continental slope which features several canyons. The deep-sea area belongs to the northern Algerian Basin (also referred to as Sardo-Balearic Basin) and exhibits water depths of up to 2800 m. According to Millot (1999), the mean surface circulation is mainly related to the inflow of "new" Modified Atlantic Water
(MAW) from the Strait of Gibraltar by means of anticyclonic eddies originating from the Algerian Current. Another branch of "old" MAW, which mixed with the underlying water masses on its large-scale cyclonic circulation through the Tyrrhenian, Ligurian, and Balearic Seas, comes probably from the west via the Balearic Current (García et al., 1996). Just below the MAW, Winter Intermediate Water (WIW) follows the path of the MAW along its whole cyclonic path. Levantine Intermediate Water (LIW) originates from the Eastern Mediterranean and the direct path to the ROPS domain is via the Sardinia Channel and then
northward around the southern tip of Sardinia. Below the LIW, Western Mediterranean Deep Water (WMDW) and Bottom Water (BW) are found.

From analyses of the REP14-MED observational data set, it turned out that the distribution of the water masses and the circulation patterns resembled the classical picture described above, but there were also significant differences. According to Knoll et al. (2017), the temperature and salinity of MAW, LIW, and BW had increased compared to the observations during
the last decade. In addition, an anticyclonic WIW eddy with unusual low temperatures and salinities was identified which may confirm the existence of a direct route of WIW from its formation region to the observational site. By contrast to previous observations, LIW occupied the whole trial area and the predominant direction of the geostrophic flow was to the north with the largest transports in the deep water off the 1000-m depth contour; no LIW vein tied closely to the Sardinian coast was found south of 40° N. The MAW pattern was different: namely, the major northward transport occurred also to the west of the
1000-m contour in a broad 30–50-km wide band but in addition, there was a narrow vein of near-coastal northward currents, the width of which rarely exceeded 10 km. Southward transport with a zonal extent of 20–40 km prevailed between the 2 northward directed regimes. Both the meridional flow bands of MAW and LIW were connected by alternating 10–30-km wide zonal currents. The observed geostrophic flow pattern suggests a mean transport to the north with superimposed mesoscale perturbations of 10–40 km in diameter. This defines another demand to ROPS to reproduce the horizontal variability of these
scales, i.e. to resolve the Rossby radius. Concerning the temporal scales, repeated ADCP (Acoustic Doppler Current Profiler) sections indicate that noticeable changes of the flow field occur within 4 days (see Fig. 14 in Knoll et al. (2017)). However, this time scale is stipulated by the minimum interval between the repeated ADCP surveys; in reality, shorter scales are likely. Hence, an additional objective is to resolve at least day-to-day changes.

The modified version of ROPS is described in Section 2. In the subsequent section is provided an overview of the observa-
tional data used in the mainframe of this article. The results of various ROPS runs are displayed in Section 4 and discussed

in Section 5. The conclusions are found in Section 6. All time specifications refer to the year 2014, and the time coordinate is UTC (Universal Time Coordinated).

## 2 ROPS

### 2.1 ROMS

The employed numerical ocean circulation model is ROMS, the Regional Ocean Modeling System. ROMS is a hydrostatic, free-surface, primitive equations ocean model, the algorithms of which are described in detail in Shchepetkin and McWilliams (2003, 2005). In the vertical, the primitive equations are discretised over variable topography using stretched terrain-following coordinates, so-called s-coordinates (Song and Haidvogel, 1994). In the mainframe of this article, spherical coordinates on a staggered Arakawa C-grid are applied in the horizontal. For the horizontal advection of momentum, a third-order upstream

bias advection scheme is used. A $4^{th}$-order, centered differences scheme is applied for the horizontal and vertical advection of tracers. The horizontal mixing of momentum and tracers is accomplished by means of a Laplacian formulation, and the vertical mixing is parameterised by the GLS (Generic Length Scale) scheme (Umlauf and Burchard, 2003) using the k-$\omega$ setup based on the turbulent closure scheme of Wilcox (1988). The air-sea interaction boundary layer in ROMS is formulated by means of the bulk parameterisation of Fairall et al. (1996). The processing of ROMS is accomplished within the grey box depicted in

Fig. 1, including nudging, data assimilation, and the proper integration.

### 2.2 The domain

While the processing of ROMS is recurring, the setup of the ROPS domain is a one-time task (Fig. 1). The domain is situated to the west of Sardinia (Fig. 2). The west and east boundaries are at 6°30.5' E and 8°35.5' E, while in the south and north the domain is limited by the 38°36.4' N and 40°59.6' N latitude circles, respectively. In east-west direction, the domain is separated

in 120 grid cells, and in south-north direction in 178 cells, which yields an average grid spacing of $\Delta x \approx \Delta y \approx 1500\,\mathrm{m}$ in the zonal and meridional direction, respectively.

    Bathymetry data from the General Bathymetric Chart of the Oceans (GEBCO) with a spatial resolution of 1 arc minute were provided by the British Oceanographic Data Centre (BODC) and mapped on the horizontal grid. Coastline data from NOAA (National Oceanic and Atmospheric Administration) were overlaid to the bathymetry and required some manual editing of

the land mask. In order to avoid crowding of the s-coordinates in shallow water regions, the bathymetry was clipped at 20 m which is the minimum allowed water depth. For the smoothing of the bathymetry, a second-order Shapiro filter was applied. After smoothing, the so-called $rx0$ parameter resulted as 0.31 which is about 50% higher than the maximum value of 0.2 recommended by Haidvogel et al. (2000), but $rx0$ is still less than 0.4 as suggested in the ROMS forum (https://www.myroms.org/forum).

In the vertical direction, the domain is separated in $K = 70$ s-layers, the position of which is controlled by 3 parameters $(\theta_s, \theta_b, h_c)$ and 2 functions, $V_{tr}, V_{str}$. Here, $V_{tr}$ is the transformation equation, $V_{str}$ the vertical stretching function, $\theta_s$ and

$\theta_b$ are the surface and bottom control parameters, and $h_c$ is the critical depth controlling the stretching (for more details, see https://www.myroms.org/wiki/). For all ROMS runs shown below, $V_{tr} = 2$, $V_{str} = 1$, $\theta_s = 5$, $\theta_b = 0.4$, $h_c = 50$ m were selected, enabling high vertical resolution near the surface. This combination of functions and parameters yielded a grid dependent parameter $rx1 = 22.7$ which is a measure for the pressure gradient error over steep topography. Namely, according to the ROMS discussion forum, $rx1 > 14$ is considered as "insane" because the Haney (1991) condition is violated, however, there are various contributions in the forum, reporting that even with $rx1 >> 14$ there did not arise any problems with the corresponding ROMS runs.

## 2.3 Initialisation and nudging

ROMS was initialised from nowcasts of the MERCATOR global ocean circulation model (Drévillon et al., 2008) via CMEMS, the Copernicus Marine Environment Monitoring Service. The downscaling from MERCATOR (the parent) to ROMS was accomplished first by linear horizontal interpolation of the prognostic fields on the ROMS grid. As the maximum horizontal resolution of the parent is 9.25 km (1/12°), the nesting ratio (also referred to as grid refinement factor) is around 6.2. In comparison with other studies applying one-way nested model setups, this ratio is rather large. For instance, Capet et al. (2008) and Gula et al. (2016) used a ratio ∼3 which is in line with the recommendation of McWilliams (2016) "Experience shows ... that the grid refinement factor should not be much larger than 3". The choice of the nesting ratio in the present article was driven by 2 criteria: on the one hand, the grid spacing should not be much larger than 1500 m to properly resolve the Rossby radius (see below); on the other hand, there were only 2 parent models available at CMEMS – MFS, the Mediterranean Forecasting System (Dombrowsky et al., 2009; Tonani et al., 2014) and MERCATOR. Namely, the higher-resolution model was MFS (∼7 km), but it was shown by Onken (2017) that initialising ROMS from MERCATOR instead of MFS provided a better agreement between the modelled fields and the observations. Moreover, precursor tests of ROMS using a grid size of 3000 km (nesting ratio ∼3.1) revealed no significant differences compared to the actual version, except for that small mesoscale features were not at all reproduced. This is in agreement with Pham et al. (2016) who demonstrated that the magnitudes of errors were comparable, using nesting rations of 3 or 6, respectively.

After downscaling, all fields were interpolated vertically from the horizontal depth levels to the s-coordinates. A special issue was the alignment of the land masks: if any wet grid cell in ROMS was covered by a dry grid cell of the parent, a smooth transition of all variables was created by taking the average of the surrounding parent cells. However, as this may lead to a violation of continuity by non-zero horizontal velocities normal to the land mask, all horizontal velocities next to the ROMS land mask were set to zero.

Later on, during the coarse of the ROMS integration, there is the option to nudge the 3-dimensional temperature and salinity fields once a day towards the parent. This guarantees that ROMS will not develop a solution in the interior of the domain which deviates significantly from the solution provided by the parent. This option is only useful if there are no data for assimilation, but in all model runs described in this article, nudging is turned off because a rich data set from observations was available (see below).

## 2.4 Lateral boundary conditions

The ROMS code includes various methods for the treatment of open boundaries. After extensive sensitivity studies, it was found out that the following algorithms served best for the posed problem: for the sea surface elevation, the Chapman condition was selected (Chapman, 1985), and for all other quantities (i.e. barotropic and baroclinic momentum, turbulent kinetic energy, eddy diffusivity), the mixed radiation-nudging conditions after Marchesiello et al. (2001) were applied.

The lateral time-dependent boundary conditions were provided as well by MERCATOR by means of one-way nesting. However, the information was not instantaneously superimposed to the ROMS solution but an additional nudging was applied to all prognostic variables which allowed these fields to adjust slowly to the parent values at the boundaries within an e-folding time scale of 2 days.

## 2.5 Surface boundary conditions

At the sea surface, boundary conditions for the air-sea exchange of fresh water, momentum, and heat were evaluated from the output of the COSMO-ME weather prediction model which was made available by the Italian Weather Service CNMCA (Centro Nazionale di Meteorologia e Climatologia Aeronautica). COSMO-ME covers the entire Mediterranean Sea with a horizontal resolution of 7 km and provides 72-hour forecasts of the wind field at 10-m height, air temperature and relative humidity at 2 m, air pressure at sea level, cloudiness, short wave radiation, and precipitation. The temporal resolution is 1 hour.

## 2.6 Data assimilation

In the ROPS runs presented below, temperature and salinity data from shipborne CTD (Conductivity-Temperature-Depth) probes and gliders were assimilated. During the integration of ROMS, OA is controlled by six parameters:

- $W$: this is the width of the time window (in hours) that determines which data are selected for assimilation. $W$ is centred around the instant $t_{assim}$ when the assimilation takes place, e.g. if $t_{assim}$=00:00 (midnight) and $W$=24 hours, data between noon of the previous day and noon of the successive day are selected.
- $C$: the correlation length scale (in km). $C$ is a 2-element vector enabling a non-isotropic Gaussian correlation for the meridional and zonal directions, respectively.
- $\delta T_{obs}$, $\delta S_{obs}$: the observational errors of temperature and salinity.
- $\delta T_b$, $\delta S_b$: the background errors of temperature and salinity.

Provided that all temperature and salinity data are stored as vertical profiles in daily directories, the data assimilation engine is invoked each day at midnight and proceeds as follows:

- The daily directories are searched for CTD profiles which fit in the desired time window $W$.
- The vertical levels are defined where the OA is performed; these levels are given by the depth of the s-coordinates at the maximum depth of the domain (Fig. 3).
- The vertical profiles are interpolated vertically on the OA vertical levels.

- As the correlation length scale $C$ is given in metric units, the ROMS spherical horizontal coordinates and the coordinates of the observations are converted to the metric Gauss-Krüger system.

- For each OA vertical level, the model prediction at the positions of the observations serves as background field for any tracer variable $\Psi$ (here: temperature $T$ and salinity $S$), and is subtracted from the observed data.

5    - OA maps the anomalies at each level on the ROMS horizontal grid and computes the normalised mapping error $\epsilon_\Psi$ at the same time.

- The background field is added to the analysed gridded fields.

- The resulting tracer fields are melded with the actual ROMS fields, using $\epsilon_\Psi$ for weighting. As $0 <= \epsilon_\Psi <= 1$, the melding for any tracer is accomplished by the algorithm

$$\Psi_{corr} = \epsilon_\Psi \Psi_{ROMS} + (1 - \epsilon_\Psi)\Psi_{obs}, \tag{1}$$

where $\Psi_{ROMS}$ is the original tracer field predicted by ROMS, $\Psi_{obs}$ are the gridded observations, and $\Psi_{corr}$ is the final corrected field resulting from the melding. Hence, if $\epsilon_\Psi$ is big (e.g. $\epsilon_\Psi = 1$ in the extreme case), no correction is applied and the ROMS solution remains unchanged. On the other extreme ($\epsilon_\Psi = 0$ if the observations are 100-% trustworthy), the ROMS solution is rejected and substituted by the observations.

## 2.7   Integration and output

All ROPS runs presented below were initialised on 1 June at 00:00 and integrated forward for 24 days until 25 June 00:00. From a precursor run, it was verified that the spin-up period was about 7 days. Hence, as the majority of observations is assimilated after 8 June, a statistical equilibrium is almost achieved at that time. To satisfy the horizontal and the vertical CFL criterion, a baroclinic time step of 108 s (800 steps per day) was chosen, and the number of barotropic time steps between each

baroclinic time step was 40. Harmonic mixing along isopycnals with an eddy diffusivity coefficient of 5 m$^2$ s$^{-1}$ was used for the horizontal diffusion of $T$ and $S$, and a viscosity coefficient of 10 m$^2$ s$^{-1}$ was selected for the diffusion of momentum. In the vertical direction, a diffusivity coefficient of $2 \times 10^{-5}$ m$^2$ s$^{-1}$ was used and the eddy viscosity coefficient was $10^{-5}$ m$^2$ s$^{-1}$. All diffusion coefficients were optimised in Onken (2017). Further on, a quadratic law using a coefficient of 0.003 was applied for the bottom friction, and the pressure gradient term was computed using the standard density Jacobian algorithm of

Shchepetkin and Williams (2001, unpublished; see `http://www.atmos.ucla.edu/~alex/ROMS/pgf1A.ps`). The 3-dimensional volume of all prognostic fields was written to an output file in 6-hour intervals.

## 3   Observational data

Observational data were selected from the REP14-MED experiment which took place 6–25 June; for a complete overview of all observations, see Onken et al. (2017a). In detail, these were

30    - 312 CTD casts taken by lowered CTD und underway CTD probes, thereof 113 on Leg 1 (6–11 June), 173 on Leg 2 (12–20 June), and 26 at the start of Leg 3 on 23 June (for the casts on Legs 1 and 2 see Fig. 4). The positions of the

casts taken during Leg 1 were arranged nominally on a 10 km×10 km grid except for 2 additional casts at 40°15'N (Fig. 4a). During Leg 2, the sampling pattern of Leg 1 was partly repeated, but extra casts were taken at the boundaries of the observational grid. Further CTD profiles close to the Sardinian coast between about 39°15'N and 39°30'N came from an acoustic experiment (Fig. 4b). The scheduled vertical extent of all casts was 1000 dbar or bottom depth (whatever was shallower) but 10 casts especially at the western boundary of the observational domain reached greater depth to characterise the deep water masses.

- 5731 CTD profiles collected by 11 gliders (Fig. 5). All gliders were deployed on 8 and 9 June, respectively, and operated until their recovery on 23 June, except for the most northern one which died on 10 June. The nominal glider tracks were arranged halfway between the zonal CTD sections (Fig. 4), thus doubling the meridional resolution of the observations. The scheduled depth of the gliders was limited by their pressure rating: 6 gliders were rated at 1000 dbar, 1 at 650 dbar, and 4 at 200 dbar.

- CTD data from ScanFish (EIVA, Skanderborg, Denmark) tows 21 June 12:03–23 June 23:38 (Fig. 6). The scheduled maximum depth of the ScanFish was around 190 m.

The temperature and salinity data from the lowered probes and from the gliders were assimilated in ROMS while the ScanFish data served for the verification of the forecasts. In Fig. 7 are shown the number of CTD profiles which were available for assimilation.

## 4  Results

In the following are presented the results of 4 series of ROPS experiments. In Series A is explored the performance of the ROPS forecasts in dependency of the correlation length scale, in Series B the sensitivity to the background errors, and in Series C the impact of the size of the assimilation window. Finally, the dependency on the forecast range is assessed in Series D.

### 4.1  The verification method

The verification of the forecast accuracy is conducted by means of root-mean-square error (RMSE) analyses which act as a metrics for the difference between the observations and the forecasts of any tracer variable $\Psi$. If there are $N$ observations and $N$ corresponding forecasts, then the squared error of the i-th observation is

$$(\Delta\Psi)^2 = [\Psi_{OBS}(x_i, y_i, z_i, t_i) - \Psi_{FC}(x_i, y_i, z_i, t_i)]^2 \tag{2}$$

where $x, y, z$ are the horizontal (eastward and northward) and vertical coordinates, respectively, $t$ is time, and the subscripts *OBS* and *FC* refer to the observations and the forecasts, respectively. The RMSE, $\Delta\Psi$, of all observations is then

$$\Delta\Psi = \sqrt{\frac{1}{N}\sum_{i=1}^{N}(\Psi_{OBS\,i} - \Psi_{FC\,i})^2} \tag{3}$$

The forecast quality is determined by the skill score $\Gamma$ which is evaluated by means of the improvement of the forecast against a reference field (Murphy, 1988),

$$\Gamma_\Psi = 1 - \frac{\Delta\Psi(FC,OBS)}{\Delta\Psi(REF,OBS)} \tag{4}$$

where $\Delta\Psi(FC,OBS)$ is the RMSE between the forecast and the observations at the forecast time $t = t_{FC}$, and $\Delta\Psi(REF,OBS)$ is the RMSE between a reference field and the observations. Here, the values of $T$, $S$, and the potential density $\sigma$ at the positions of the observations and at the instant $t = t_{INI}$ when the forecast was initialised, are serving as reference (persistence assumption). Hence, a perfect forecast would yield $\Gamma_\Psi = 1$ because the forecast agrees exactly with the observations and $\Delta\Psi(FC,OBS) = 0$. A successful or good forecast would mean $\Delta\Psi(FC,OBS) < \Delta\Psi(REF,OBS)$ and $0 \leq \Gamma_\Psi \leq 1$ because the forecast is closer to the observations than the reference ("forecast beats persistence"). By contrast, $\Gamma_\Psi \leq 0$ would be a criterion for an unsuccessful of bad forecasts because $\Delta\Psi(FC,OBS) > \Delta\Psi(REF,OBS)$. In the following, $\Gamma_\Psi$ is applied both to single s-layers and to the mean

$$\overline{\Gamma_\Psi} = \frac{1}{s_2 - s_1 + 1} \sum_{s_1}^{s_2} \left[ 1 - \frac{\Delta\Psi_{wgt}(FC,OBS)}{\Delta\Psi_{wgt}(REF,OBS)} \right] \tag{5}$$

which is the average over all s-layers from s-layer no. $s_1$ to s-layer no. $s_2$. The subscript $wgt$ indicates weighting by the layer thickness in order to take account of the different masses of each layer.

In all ROPS runs presented below, the data from the ScanFish survey were utilised for verification. As the survey was completed within about 60 hours, it was considered to be synoptic and centred at $t = t_{VER} =$ 22 June 18:00. $\Delta\Psi$ and $\Gamma_\Psi$ were evaluated at the same instant, hence $t_{VER} = t_{FC}$ and the time dependency in (2) was removed. The synopticity assumption was somwhat risky because the expected scales of the temporal variability were less than 4 days (see Introduction). However, assuming non-synoptic conditions would have required to interpolate the ROMS tracer output in 3-dimensional space and time on each observation, or vice versa to interpolate each observation on the ROMS grid – any of these actions would have been too expensive. Moreover, none of them was mandatory because the results shown below are consistent and conclusive. In order to make the ScanFish observations suitable for a comparison with the ROMS model output, the trajectories were hacked in 629 upward and downward profiles, and a mean time and a mean position were assigned to each profile. All temperature and salinity profiles were mapped with OA on constant depth levels on the ROMS horizontal grid, using a correlation scale $C = 1.8$ km. Thus, as the along-track distance between the individual profiles was 500–700 m, 3 to 4 observations were contributing significantly to the mapping at each horizontal grid point. Finally, the analysed fields were interpolated from the horizontal OA levels on the ROMS vertical grid.

The observational errors for temperature and salinity, $\delta T_{obs}$ and $\delta S_{obs}$, respectively, were determined from the standard deviation of the respective fields on each OA level. A special problem arose for the determination of the background errors: usually, one would compute these errors from the standard deviation of the background field, but in this special case the background was the mean of the observations (a single number), and the standard deviation would be zero. Therefore, they were defined as $\delta T_b = 5 \cdot \delta T_{obs}$ and $\delta S_b = 5 \cdot \delta S_{obs}$ which pushed the analysed fields as close as possible to the observations. Fig. 8 illustrates the result of this procedure using the example of the ScanFish section A09 (cf. Fig. 6). The analysed fields in Fig.

8d, e, f resemble almost perfectly the observations shown in Fig. 8a, b, c. Later on, for the evaluation of the forecast accuracy and the skill score, $\Delta\Psi$ in Eqs. (3) and (4) was multiplied by $(1 - \epsilon_\Psi)$. As $\epsilon_\Psi = 0$ at the exact position of the observation and $0 < \epsilon_\Psi \leq 1$ elsewhere, $\Delta\Psi$ became significantly different from zero only in the immediate vicinity of the observation.

## 4.2   Series A: the impact of the correlation length scale

The natural correlation scale is the internal Rossby radius which in the western Mediterranean Sea lies between 3 and 13 km for the second and the first mode, respectively, depending on the season (Grilli and Pinardi (1998), Robinson et al. (2001)). For OA, however, one must not uncritically select any number within this range for $C$ because this could have unpleasant side effects: if $C$ would be significantly less than the mean horizontal distance between the observations, then OA would create unrealistic eddy-like features centred at the sites of the observations. On the other extreme, realistic mesoscale and sub-mesoscale structures would be blurred if $C$ were significantly greater than the Rossby radius. While the horizontal distribution of the shipborne CTD casts was isotropic (mean distance 10 km), the glider CTD data were strongly anisotropic: the mean meridional distance between the glider tracks was also about 10 km, but the zonal resolution was $\mathcal{O}(100 \text{ m})$ in shallow water and $\mathcal{O}(1000 \text{ m})$ in deeper water. In this series, 8 ROPS runs with different assumptions for the correlation length scale $C$ were conducted. $C$ was selected isotropic because a preliminary processing of data from shipborne ADCPs had revealed that the major part of the model domain was characterised by an eddy field with alternating currents; only along the west coast of Sardinia, predominantly meridional currents were prevailing in a $\approx$10-km wide stripe. The selected values for $C$ were 2.5, 5.0, 7.5, 10.0, 12.5, 15.0, 17.5, and 20 km, respectively.

In the A-Series, all CTD and glider data which were collected until $t_{INI}$=18 June 00:00 were assimilated. The size of the assimilation window was $W = 24$ hours. The observational errors were set to fixed values $\delta T_{obs} = 1.3°$ C and $\delta S_{obs} = 0.2$ in all OA layers; these were the maximum values of the respective standard deviations found in the upper thermocline. In precursor tests, $\delta\Psi_{obs}$ was set to the standard deviation of $\Psi$ at the respective OA level (as was done for the OA of the ScanFish observations, see above), but here this strategy failed because in the deeper layers the standard deviation was approaching zero due to the horizontal homogeneity of the water body, and the OA package generated unrealistic solutions which caused ROMS to blow up shortly after the instant when data were assimilated. For similar reasons, $\delta\Psi_b$ was not derived from the standard deviation of the background field because the isotherms and isohalines in the deep ocean were almost horizontal which originated from the MERCATOR solution. Therefore, $\delta\Psi_b = \delta\Psi_{obs}$ or $r_\Psi = \delta\Psi_b/\delta\Psi_{obs} = 1$ was selected as a first guess. This was a rather conservative approach but it enabled the OA to find the optimum solution about halfway between the observations and the background fields. After the last assimilation on 18 June, ROMS was integrated forward in a free mode, i.e. it was no more constrained by observations. Finally, the model results were verified against the ScanFish observations at $t_{FC}$=22 June 18:00. For an overview of the parameter settings and results, see Table 1.

In Fig. 9 are shown the vertical distributions of $\Delta T$, $\Delta S$, and $\Delta\sigma$ for ROPS runs A1–A3, and A5–A8 (Run A4 is missing; it died on 14 June shortly after midnight, apparently because ROMS could not cope with the density field created by the assimilation). These quantities are evaluated in the ROMS vertical layers and plotted vs. the layer number, starting with layer 1 at the seabed. The graphs are empty for layers 1–9 and 69–70 next to the sea surface because the corresponding depth

ranges were never reached by the ScanFish. In order to have an objective measure which correlation scale provided the best forecast, $\Delta\Psi$ was averaged over all layers. The resulting layer thickness-weighted mean values $\overline{\Delta T}$, $\overline{\Delta S}$, and $\overline{\Delta\sigma}$ are written in the rightmost column of the legend of the graph and in Table 1 as well. Generally, $\Delta\Psi$ is decreasing from the surface to greater depth, however, rather low values are found in layer 68 which covers the vertical range between about 7 m at the

5 maximum depth of the domain (cf. Fig. 3) and 0.6 m in the shallowest regions. This layer is characteristic for the mixed-layer, the properties of which are controlled by the larger scale uniform weather patterns. The maxima below in layer 49 are caused by the higher spatial variability in the thermocline as this layer ranges from about 10 to 220-m depth. $\overline{\Delta T}$ lies between $2.74 \times 10^{-3}\,^\circ$C in A7 and $3.11 \times 10^{-3}\,^\circ$C in A2 but the variance among all runs is rather small. For $\overline{\Delta S}$, the minimum of $6.20 \times 10^{-4}$ is found in A3 and the maximum of $8.80 \times 10^{-4}$ in A1. $\overline{\Delta\sigma}$ is minimum in A8 ($5.59 \times 10^{-4}$ kg m$^{-3}$) and the

10 maximum of $8.68 \times 10^{-4}$ kg m$^{-3}$ is attained in A1. Hence, for $\overline{\Delta\sigma}$, there is a clear tendency that an increase of the OA correlation length scale appears to improve the accuracy of the forecast. Similar tendencies may be seen for $\overline{\Delta T}$ and $\overline{\Delta S}$.

The vertical distributions of the skill scores $\Gamma_\Psi$ and the corresponding layer weighted means are displayed in Fig. 10. Positive scores indicating a successful forecast of temperature were obtained in all runs (except for A4 which died), and the maximum of $\overline{\Gamma_T} = 27.0\%$ was attained in A5. For salinity, only the A3 and A5 forecasts beat persistence but with a rather low score of

15 only 4.3 and 0.2%, respectively. $\overline{\Gamma_\sigma}$ was positive for runs A1 and A3–A8, and the highest score of 26.4% was achieved in A5 for $C = 12.5$ km. This is remarkably in line with Grilli and Pinardi (1998) who found the first mode Rossby radius between about 11 and 13 km in the waters to the west of Sardinia.

Compared to the RMSE analysis above, the mean skill scores do not exhibit any correlation scale-dependent trend. Instead, there are maxima of $\overline{\Gamma_T}$ in A5, $\overline{\Gamma_S}$ in A3, and $\overline{\Gamma_\sigma}$ in A5, and the scores decrease both for smaller and larger correlation scales.

This potentially contradictory behaviour needs an explanation: $\Delta\Psi$ is a measure for the accuracy of the forecast which is evaluated from the forecast and the observations on 22 June 18:00 at the locations of the observations, cf. eq. (3). The decrease of $\Delta\Psi$ with increasing $C$ means that the forecasts using larger correlation scales for the generation of the initial conditions at $t = t_{INI}$ are closer to the observations than those forecasts using smaller scales, irrespective of the initial conditions themselves. Presumably, the larger correlation scales create already initial conditions which are rather close to the observations.

This is illustrated by Fig. 11: there are shown $\Delta\sigma(REF, OBS)$ for A1 and A8, where the potential density fields at $t = t_{INI}$ served as reference. Evidently, everywhere above layer 30, $\Delta\sigma$ in A8 ($C = 20$ km) is much closer to the observations than $\Delta\sigma$ in A1 using $C = 2.5$ km. By contrast, $\Gamma_\Psi$ is a measure of the improvement of the forecast with respect to the reference, and it simply states that the highest forecast quality is obtained if the horizontal wavenumber spectrum of the initial conditions is peaked at the Rossby radius. Therefore, the A5 forecast using $C = 12.5$ km was considered as the best of the A-Series because

of the high skill score for potential density and served as control run in the following B-Series. In addition, in all ROPS runs discussed below, $\Delta\Psi$ was not any longer utilised as a criterion for the forecast skill score.

### 4.3 Series B: the impact of background errors

In Series B, the dependency of $\Gamma_\Psi$ on $\delta\Psi_b$ was investigated while $\delta\Psi_{obs}$ was kept constant. Eight different configurations B1–B8 were tested using $r_\Psi = \delta\Psi_b/\delta\Psi_{obs} \in \{.1, .5, 1, 2, 3, 4, 5, 6\}$. As $r_\Psi$ was continuously increasing with increasing sequence

number, the weighting of the background field decreased at the same time and the objectively analysed temperature and salinity were forced closer to the observations. Note that B3 was the control run identical with A5.

As can been from Fig. 12, the mean forecast skill for temperature was positive for all runs and the maximum of $\overline{\Gamma_T} = 28.2\%$ was attained in B2 using a ratio $r_\Psi = 0.5$. Thus, a background error being half the observational error produced the best forecast. For $r_\Psi = 0.1$, $\overline{\Gamma_T}$ dropped suddenly to 5.8 % in B1 but increasing $r_\Psi$ from 0.5 to 6.0 in B8 caused a rather smooth decrease from the maximum in B2 to 14.2 % in B8. For salinity, $\overline{\Gamma_S}$ was mostly negative or close to zero, and the best skill score of 1.4 % was obtained in B8 for $r_\Psi = 6.0$. Despite the negative score for salinity, $\overline{\Gamma_\sigma}$ was always positive except for Run B6; the highest score of 26.4 % was recorded in B3 using $r_\Psi = 1$. Therefore, B3 served as control run in the subsequent C-Series.

## 4.4   Series C: the impact of the assimilation window

In all previous runs, the data assimilation engine was invoked each day at 00:00 hours. As the size of the assimilation window was $W = 24$ hours, observational data between noon of the previous day and noon of the actual day were assimilated. This setting for $W$ was the minimum because smaller values would lead to non-consideration of data. In this subsection, the impact of larger windows $W$ on the skill score is investigated in 5 ROPS runs C1–C5, applying $W \in \{24, 30, 36, 42, 48\}$ hours where C1 is the control run identical with B3. However, C4 and C5 using a windows size of 42 and 48 hours, respectively, blew up on 15 June. Obviously, very large windows were not suitable because the actual ROMS forecast was blended with too old observational data and with data which lay too far in the future. This is in line with the Introduction where it was stated that the expected time scales of the temporal variability were less than 4 days. One may argue that a few more CTD profiles must not have lead to a model crash, but one has to consider that the gliders provided up to more than 400 profiles every day (see Fig. 7), and an extension of $W$ by just six hours would mean that about 100 additional profiles which were too much decorrelated in time with the actual forecast, would contribute to the assimilation fields. The skill scores of the remaining runs C1–C3 are displayed in Table 1: the best score for $\overline{\Gamma_\sigma}$ was again reached in the control Run C1 but also in C2 and C3, the scores were higher than 20 %. Worth mentioning are the positive but rather small scores for $\overline{\Gamma_S}$ in C2 and C3. Anyway, because of the maximum scores for $\overline{\Gamma_t}$ and $\overline{\Gamma_\sigma}$, C1 was selected as control run in the following Series D.

## 4.5   Series D: the impact of the forecast range

In this series, 12 ROPS runs D1–D12 with different forecast ranges were conducted and verified as before. In all runs, the parameter settings of C1 were utilised but the initialisation time $t_{INI}$, i.e. the time when the last data assimilation took place, was varied between 11 and 22 June. In D1, CTD data were assimilated until 11 June 00:00. Thereafter, ROMS was integrated forward in a free mode, i.e. it was no more constrained by observations. The forecast range $\tau$ was the time span between the instant when the last assimilation took place and the verification time $t_{FC} = 22$ June 18:00, thus 11.75 days. In D2, the last data were assimilated on 12 June, in D3 on 13 June and so on. Hence, in runs D2–D12, $t_{INI}$ was advanced by 24 hours in each case until $t_{INI} = 22$ June 00:00 in D12, and correspondingly the forecast range shrunk progressively in 1-day steps from $\tau =$11.75 days in D1 to $\tau =$0.75 days in D12.

The skill scores of all runs in dependency of $t_{INI}$ and $\tau$ are summarised in Table 2, and in Fig. 13 are shown the graphs of $\overline{\Gamma_T}$, $\overline{\Gamma_S}$, and $\overline{\Gamma_\sigma}$. For D1 ($t_{INI} = 11$ June, $\tau =$ 11.75 days), $\overline{\Gamma_\sigma}$ attained the absolute maximum of 31.9 % within this series (Fig. 13c). Skill scores around 30 % were also reached in D2 and D3. In D4–D12 towards smaller forecast ranges, the score exhibited an overall decreasing trend but it remained positive except for D11 where $\overline{\Gamma_\sigma} = -22.4$ %. The characteristics of the

$\overline{\Gamma_T}$ curve resembles closely that of $\overline{\Gamma_\sigma}$. In terms of qualitive arguments, these are the high scores in D1–D3, and the decrease afterwards. Quantitatively, these are the scores around or even above 30 % in D1–D3, the moderate values around and below 10 % in D5 and D6, the scores above 25 % at the relative maximum in D8, the minima in D11, and the recovery to positive values in D12. The $\overline{\Gamma_S}$ curve is correlated with the graphs of $\overline{\Gamma_\sigma}$ and $\overline{\Gamma_T}$, concerning the overall decreasing trend and the locations of the relative minima and maxima. However, the skill scores for salinity are always lower than those for density and temperature

in D1–D9 and D12, frequently even being negative. The highest values above 10 % are attained in D1 and D2 – a rather modest score compared to the $\approx$30 % scores of $T$ and $\sigma$ at the same time.

In order to assess the impact of the data assimilation as a whole, another ROPS run was conducted referred to as D0. This run was identical to all other runs of the D-Series but no data were assimilated at any time. For D0, the skill scores were computed in the same way as for D1–D12 for each initialisation time day between 11 and 22 June, and in addition for "virtual"

initialisations on 1–10 June. The corresponding curves (the thin lines) are overlain to the graphs of $\overline{\Gamma_\Psi}$ in Fig. 13a, b, c. The skill scores of D0 are positive for the majority of the initilisation times $t_{INI}$. Negative values for $\overline{\Gamma_T}$ are only obtained for $t_{INI} \in \{1, 2, 3\}$ June, for $\overline{\Gamma_S}$ and $t_{INI} \in \{15, 17, 18, 19, 21\}$ June, and for $\overline{\Gamma_\sigma}$ and $t_{INI} \in \{19, 21\}$ June. Thus, although no data were assimilated in D0, the forecasts beat persistence in most cases for forecast ranges of at least 3 weeks. Other particular feature of the D0 forecasts are the maximum skill score for $t_{INI} = 8$ June and the decreasing trend thereafter. Except for D6

($t_{INI} = 16$ June) and 20 June $<= t_{INI} <= 22$ June, the skill scores $\overline{\Gamma_\sigma}$ of D0 are always lower than the corresponding scores of D1–D12; hence, the assimilation of observational data has definitely improved the forecast quality for potential density. About the same proposition is valid for $\overline{\Gamma_T}$ but not for $\overline{\Gamma_S}$: here, except for $t_{INI} \in \{17, 18, 19\}$ June, the skill score of D0 is always higher than in D7–D9. This strange behaviour – and as well some other possibly weird findings in this subsection – need explanations which will be given in the Discussion below.

**5  Discussion**

The first objective of this article was to demonstrate that ROPS produces good forecasts. Murphy (1993) defines 3 types of "goodness": consistency, quality, and value. Concerning the latter, it is rather difficult to rate the value for the "end users" because REP14-MED was planned solely for scientific purposes – see the objectives defined in Onken et al. (2017a). Amongst others, a special aim was the comparison of different methods for data assimilation. This article is the third one within a

series of 5 whereof Oddo et al. (2016) and Onken (2017) are published. Two articles using the same observational data are in preparation, applying the Ensemble Kalman Filter and 4D-Var assimilation methods. Hence, the value of the ROPS forecasts may be judged at the time when all papers are published. The consistency of the ROPS forecasts was assessed by comparison with the observations described by Knoll et al. (2017), using the output of ROPS run C1 on 20 June. In detail, the large-scale

horizontal distributions of $T$ and $S$ at 50 and 400-m depth (these are the depths of the MAW and LIW cores, respectively) resembled the observed patterns, but the contours were shifted against each other by several miles. This was plausible because the observed fields were averaged over the observational period 8–18 June while the forecast was a snapshot. The same applied for the predicted currents which were checked against the observed geostrophic transports. It was also verified that the data

assimilation did not create any unrealistic water masses in those regions where nearby observations were available. In order to assess the impact of the rather large nesting ratio, the vertical velocity along the lateral boundaries was frequently checked for strange patterns, but no abnormal behaviour was detected at any time. This was not surprising, because to minimise false advection effects, the distance between the open boundaries and the observations was 30 miles in the west and 45 miles in the south and the north, respectively (cf. Figs. 4, 5).

With respect to the forecast quality, a major result found above was that the mean skill scores $\overline{\Gamma_T}$, $\overline{\Gamma_S}$, and $\overline{\Gamma_\sigma}$ decreased concurrently with a decreasing forecast range $\tau$. As this feature was observed both for the assimilation runs D1–D12 and for the free run D0, it can be excluded that it was somehow caused by the assimilation of observational data. Therefore, the components of the equation which determines the skill score were investigated. In particular, a closer look was taken at $\Delta\Psi_{wgt}(FC,OBS)$ and $\Delta\Psi_{wgt}(REF,OBS)$ in eq. (5). However, as these expressions represent the weighted RMSE of each individual s-layer, it

is not possible to relate them to the mean skill score $\overline{\Gamma_\Psi}$. Therefore, only for the purpose of the discussion, the mean skill score was re-defined as

$$\overline{\Gamma_\Psi^*} = 1 - \frac{\overline{\Delta\Psi_{wgt}}(FC,OBS)}{\overline{\Delta\Psi_{wgt}}(REF,OBS)} \tag{6}$$

Here, $\overline{\Gamma_\Psi^*}$ is the mean skill score computed from the mean layer RMSEs while $\overline{\Gamma_\Psi}$ as defined in eq. (5) is the mean score computed from the individual layer RMSEs. In Fig. 14 are shown $\overline{\Delta\sigma_{wgt}}(FC,OBS)$, $\overline{\Delta\sigma_{wgt}}(REF,OBS)$, and $\overline{\Gamma_\sigma^*}$ for D1–

D12 and for D0, in dependency of the forecast range $\tau$ (bottom axis) and simultaneously of the initialisation time $t_{INI}$ (top axis). First of all, $\overline{\Gamma_\sigma^*}$ and $\overline{\Gamma_\sigma}$ in D1–D12 (compare Figs. 14b, 13c) are almost identical which legitimates the re-definition in eq. (6). By contrast, the shape of the corresponding graphs for the no-assimilation run D0 differ from each other: the $\overline{\Gamma_\sigma^*}$ curve is smoother than the one of $\overline{\Gamma_\sigma}$, but the increasing trend for 1 June $< t_{INI} <$ 8 June and the decreasing trend thereafter are reproduced which is important for this discussion. According to Fig. 14a, $\overline{\Delta\sigma_{wgt}}(FC,OBS)$ (thin red line) is constant

for all initialisation times $t_{INI}$. This is trivial because the RMSE between the forecasted fields and the observations on 22 June never change, regardless of the virtual initialisation time. This facilitates the discussion because the skill score depends now solely on $\overline{\Delta\sigma_{wgt}}(INI,OBS)$ (thin blue line). The shape of the graph of the latter is identical with the shape of the $\overline{\Gamma_\sigma^*}$ curve which means that for all initialisation times $t_{INI} >$ 8 June, the ROMS initial fields are progressively approaching the verification fields with increasing $t_{INI}$. Apparently, some unknown process or the combination of different processes is driving

the model already towards the future observations without data assimilation. Potential candidates could be the downscaling of the MERCATOR fields on 1 June (see above, Section 2.3) enabling a more realistic circulation pattern, the MERCATOR forcing at the lateral boundaries, or the daily updated COSMO-ME forecasts which would not be available in real operational conditions. The opposite is the case for $t_{INI} <$ 8 June: here, the ROMS initial fields deviate progressively from the verification

fields with increasing $t_{INI}$. Probably, ROMS needs a certain spin-up time to equilibrate all fields which would be about 8 days in the present situation.

For the assimilation runs D1–D12, $\overline{\Delta\sigma}_{wgt}(FC, OBS)$ (Fig. 14a, bold red line with dots) is decreasing continuously with increasing $t_{INI}$. Hence, the later ROMS switches to the free mode without data assimilation, the closer is the forecast to the observations. This is not trivial because each assimilation cycle could create "assimilation shocks" and mess up the model dynamics (Evensen, 2003; Counillon et al., 2016). Probably, this happened in D1–D5 where $\overline{\Delta\sigma}_{wgt}(FC, OBS)$ is at about the same level as the corresponding quantity in D0, but in D6–D12 (16 June $<= t_{INI} <=$ 22 June), $\overline{\Delta\sigma}_{wgt}(FC, OBS)$ is below the horizontal line which indicates that the predicted density pattern is closer to the observations than in the no-assimilation run. This does not necessarily mean that the skill score is higher, because $\overline{\Gamma_\sigma^*}$ depends on the ratio $\overline{\Delta\sigma}_{wgt}(FC, OBS)/\overline{\Delta\sigma}_{wgt}(REF, OBS)$ according to eq. (6). As shown by the the bold blue line with dots, the denominator is mostly greater than the numerator (except for D11), and also its overall slope is larger. Consequently, the ratio is mostly $< 1$ leading to a positive skill score. Moreover, in D1–D4 and D7–D8, the ratio is small and correspondingly, the skill score is large. By contrast in D5, D6, D9, D10, and D12, the ratio is close to 1 and the skill score is approaching zero. This effect controls also the overall negative trend of the skill score because the numerator and the denominator are approaching each other with decreasing forecast range $\tau$. In other words, if the forecast range is small, then the reference fields are already very close to the verification fields, and no significant improvement can be achieved by further forward integration of the numerical model.

In all runs shown above, except for D10 and D11 (see Tables 1 and 2), $\overline{\Gamma_T}$ was greater than $\overline{\Gamma_S}$, frequently even much greater. It can be excluded that this was due to an error in the OA or in the melding procedure (eq. (1)), as the same subroutines were used for $T$ and $S$. Additional evidence was found from D0: Figs. 13a, b clearly show that $\overline{\Gamma_S}$ was always less than $\overline{\Gamma_T}$, at least for 14 June $<= t_{INI} <=$ 22 June. As the OA or eq. (1) were never applied in D0, neither could be the cause for this weird behaviour. Likewise, errors during the processing of the data for assimilation can be precluded. Another possible source of error could be the computation of the forecast skills. However, the coding of eqs. (2)–(5) was checked several times and no error was detected. Hence, it is concluded that some physical process is not properly parameterised in ROMS which induces the different skill scores of $T$ and $S$. The only process which came to the mind of the author is double diffusion which effectuates higher vertical diffusivities for salinity than for temperature (Schmitt, 1981). As shown by Zhang et al. (1998), the consideration of double-diffusive mixing in a general circulation model can have a significant impact on the horizontal transport of heat and salt, even is a conservative approach is applied to the parameterisation. The stratification in the Mediterranean Sea is especially favourable for double diffusion because of the high-salinity core of Levantine Intermediate Water below the main thermocline (Millot, 1999). Onken and Brambilla (2003) have shown that in the Algerian Basin and below about 300 m depth, the vertical diffusivity of salt may be up to twice as high as the diffusivity of heat. This leads to an enhanced diffusion of buoyancy and similarly may affect the entire circulation pattern and the different skill scores for temperature and salinity.

During the last 2 decades, the forecast skill of operational models was verified against observations in an increasing number of articles, the majority of which aimed at global models. In an early paper, Smedstad et al. (2003) showed that the skill score of the NLOM (Naval Research Laboratory Layered Ocean Model) forecast model decreased with increasing forecast range. However, the skill score was not evaluated against persistence but against climatology, both for the global domain and for a

subdomain in the Gulf Stream region. In the latter, the skill score decreased at a faster rate than in the global domain which indicated the reduced predictability in the more energetic regions. Ten years later in the mainframe of GODAE (Global Ocean Data Assimilation Experiment, Bell et al. (2009)), Lellouche et al. (2013) computed the skill score of the sea level anomaly forecast for different setups of the MERCATOR global model against persistence. They demonstrated that for most regions

the skill score was positive, except for the North Atlantic, the Mediterranean Sea, and Antarctica. Also in the mainframe of GODAE, Ryan et al. (2015) verified six different forecasting systems against climatology and persistence. Their main results were that the climatology skill score of all systems was positive for all tested parameters, while the persistence skill score (PSS) was partly close to zero or even negative for short forecast ranges, both for temperature and salinity. Afterwards, however, the PSS increased with longer forecast ranges of up to 5 days. Moreover, the PSS of salinity was mostly lower than the PSS of

temperature. Forecast skill assessments of regional models were conducted by various authors. Tonani et al. (2009) evaluated the forecast skill score of the Mediterranean Forecasting System (MFS, Pinardi (2003), horizontal resolution $\approx 7$ km) by means of comparisons with observational data from moorings, ARGO floats, and XBT (expendable bathythermograph) casts at 3 different vertical levels. It was demonstrated that the skill scores of temperature and salinity increased with increasing forecast range, reaching a maximum of about 45% for temperature around the $6^{th}$ day of the forecast. This is about 30% higher

than the maximum skill score $\Gamma_T$ determined above (see Section 4.5) which was 34% for $t_{INI} = 13$ June, corresponding to a forecast range of about 10 days. For the very short forecast range of 2 days, the skill scores were negative, and right at the surface and in the upper thermocline, the skill score for salinity was mostly lower than for temperature. A generally lower skill score for salinity, which is in agreement with the results of this article, was found as well by Chiggiato and Oddo (2008) for 2 higher-resolution operational models of the Adriatic Sea. Tonani et al. (2009) evaluated also the components of the skill score.

They demonstrated that $\Delta\Psi(FC, OBS)$ and $\Delta\Psi(REF, OBS)$ both for $\Psi \equiv T$ and $\Psi \equiv S$ were approaching each other with increasing forecast range; this is comparable to the findings from Fig. 14. On the whole, the sometimes surprising results of this article are in line with other publications.

    It was demonstrated that good forecasts can be obtained from a prediction system using OA for assimilation. The ROPS runs of the D-Series have shown that the assimilation of CTD data leads to an increase of the skill score for temperature and density,

except for those runs with a rather short forecast range of less than 3 days, e.g. D10–D12. Here, the forecast quality of the no-assimilation run D0 is superior. Most likely, it is the massive amount of assimilation data which disequilibrates the terms in the governing equations of ROMS, and a few days are required to restore the equilibrium. However, this does not imply that the accuracy of the forecast is becoming worse at the same time. This is impressively demonstrated in Fig. 14a which shows that the RMSE between the forecast and the verification (i.e. the bold red curve with dots) is monotonically decreasing

with a decreasing forecast range. Furthermore, it was shown that a vast number of observational data can be managed by OA without data reduction. In the mainframe of this article, 6034 CTD profiles were available for assimilation 7–23 June; hence, 377 profiles were assimilated every day at midnight on average. It would be worth to explore whether this potential oversampling leads to an improvement or even a deterioration of the forecast quality, compared to ROPS runs where less data would be assimilated. Though, this would first require the development of a meaningful methology for data reduction. Different

approaches for the reduction of observational data could be utilised to address quite a number of interesting questions, such as

– Are deep CTD casts needed to improve the forecast skill scores? Several deep casts extending to more than 2500 m depth were taken at the western boundary of the observational domain to assess the hydrography of the deep water masses.

– What is the impact of "deep" gliders on the skill score? During REP14-MED, 4 gliders had a pressure rating of 200 dbar, 1 was rated to 650 dbar, and 6 gliders took samples down to 1000 dbar. The impact of the deep gliders could easily be assessed if their profiles were clipped at 200 m.

– What is the cost/benefit ratio of adding more gliders? Eleven gliders were operating on 10 zonal tracks (see Fig. 5). As the most northern one died early, there were still 9 tracks G02–G10 occupied continuously for more than 2 weeks. In the first run of a cost/benefit analysis, only the data of the glider on the central track G06 would be assimilated, in the second run data from tracks G02 and G10 would be added, the third run would assimilate data from tracks G02, G04, G06, G08, and G10, and in the final run, data from all tracks would be used. Thus, the meridional resolution would be doubled during consecutive runs, and the skill score versus the resolution could be assessed.

However, to find answers to these questions is beyond the scope of this article and might be addressed in a follow-up paper.

Finally, it was demonstrated that ROPS is able to provide timely forecasts on a commercially available Personal Computer. All ROPS runs were conducted on a DELL Precision Tower 7910 using 4 processors. The CPU time of D12 which performed the maximum of 16 assimilation cycles was 4.8 hours while it was only 2.7 hours for D0 without any assimilation. Hence, as ROPS was integrated over 24 days, the CPU time for solving the primitive equations was just 6.8 minutes per day. In case that on average 377 CTD profiles were assimilated, the CPU time nearly triplicated to 18 minutes. This rather modest increase was effectuated by the pre-selection of observational data in daily directories which considered only those data for assimilation which fitted in the time window $W$. The triplicating of CPU time is still a reasonable figure compared to operational models employing 4D-VAR where the CPU time may increase by at least one order of magnitude.

## 6   Conclusions

The Relocatable Ocean Prediction System (ROPS) was employed in hindcast mode to a huge data set which was collected in in June 2014. Using objective analysis (OA), the observational data were assimilated, and the ROPS forecasts were verified against independent data.

The OA is controlled by 4 parameters which are $C$: the correlation length scale, $r_\Psi$: the ratio of background and observational errors for temperature and salinity, respectively, and $W$: the width of the time window where data are assimilated. Sensitivity tests to variations of these parameters were conducted by means of various ROPS runs encompassing the period 1–24 June. Observational data were assimilated 7–18 June, and the forecasts were varified against the verification data set on 22 June. The highest skill scores were obtained for $C = 12.5$ km, $r_\Psi = 1$, and $W = 24$ hours.

Additional runs revealed a decreasing tendency of the skill score with decreasing forecast range, where the forecast range was the time span between the verification time and the instant when the last assimilation took place. The same tendency was exhibited by a control run without assimilation which excludes the OA from being responsible for this behaviour. A

thorough analysis of the terms in the equation, which determines the skill score, revealed that persistence is approached steadily for continuously decreasing forecast ranges, and the late assimilation of observational data cannot any longer effectuate a significant improvement of the forecast skill. For extremely small forecast ranges, the skill score even became negative, because the assimilation disequilibrated the balance of forces in the dynamical model.

In all ROPS runs, including the run without assimilation, the skill score for temperature was mostly higher than the corresponding score for salinity. This is in agreement with other research papers, and it is speculated that this mismatch is due to double-diffusive processes which were not adequately parameterised.

    ROPS is able to provide timely forecasts even on commercially available Personal Computers.

*Author contributions.* N/A

10 **Code availability**

All work related to this article was done on a Linux workstation under Kubuntu 16.04. ROMS/TOMS version 3.6 was used for the model runs, the pre- and postprocessing was done with MATLAB R2016b, and the article was written in LaTeX. The model code and all scripts are available from the author on request.

**Data availability**

All data of the REP14-MED experiment are available on the CMRE data server at http://geos3.cmre.nato.int/REP14, last access 10 March 2017. The data are NATO UNCLASSIFIED and available only for the partners of the experiment. However, interested institutions can sign up for partnership at any time. Requests may be directed to <geos-webmaster@nurc.nato.int>.

**Competing interests**

The author declares that he has no conflict of interest.

*Acknowledgements.* The author would like to thank the masters and crews of NRV Alliance and RV Planet for their professionalism during the conduction of the experiments at sea. The data from COSMO-ME were provided by the Italian weather service Centro Nazionale di Meteorologia e Climatologia Aeronautica (CNMCA) and the MERCATOR data sets were downloaded from the Copernicus Marine Environment Service (CMEMS). Bathymetry data from the General Bathymetric Chart of the Oceans (GEBCO) originated from the British Oceanographic Data Centre (BODC), and coastline data were obtained from the National Oceanic and Atmospheric Administration (NOAA). 25 The author is grateful to 2 anonymous reviewers who helped to improve the article. REP14-MED was sponsored by HQ Supreme Allied Command Transformation (Norfolk, VA, USA).

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

**Table 1.** Parameter settings and results of ROPS runs in Series A,B, C. Bold numbers indicate those parameters which are varied within the respective series. The best run of each series is marked by an asterisk and serves as the control run for the successive series. Runs which blew up are marked by the † symbol.

| Run | $C$ | $r_\Psi$ | $W$ | $\overline{\Delta T}$ | $\overline{\Delta S}$ | $\overline{\Delta\sigma}$ | $\overline{\Gamma_T}$ | $\overline{\Gamma_S}$ | $\overline{\Gamma_\sigma}$ |
|---|---|---|---|---|---|---|---|---|---|
| | [km] | | [hours] | $[10^{-3}\,^\circ\mathrm{C}]$ | $[10^{-4}]$ | $[10^{-4}]$ | [%] | [%] | [%] |
| Series A | | | | | | | | | |
| A1 | **2.5** | 1.0 | 24 | 3.04 | 8.80 | 8.68 | 2.4 | -10.0 | 2.5 |
| A2 | **5.0** | 1.0 | 24 | 3.11 | 7.45 | 7.94 | 3.0 | -18.9 | -1.0 |
| A3 | **7.5** | 1.0 | 24 | 2.99 | 6.20 | 8.08 | 15.6 | 4.3 | 20.5 |
| A4† | **10.0** | 1.0 | 24 | – | – | – | – | – | – |
| A5* | **12.5** | 1.0 | 24 | 2.96 | 6.43 | 6.98 | 27.0 | 0.2 | 26.4 |
| A6 | **15.0** | 1.0 | 24 | 3.04 | 7.06 | 6.18 | 12.8 | -12.9 | 22.1 |
| A7 | **17.5** | 1.0 | 24 | 2.74 | 6.63 | 6.01 | 11.8 | -9.1 | 14.9 |
| A8 | **20.0** | 1.0 | 24 | 2.83 | 7.21 | 5.59 | 4.5 | -17.7 | 22.9 |
| Series B | | | | | | | | | |
| B1 | 12.5 | **0.1** | 24 | – | – | – | 5.8 | -7.5 | 5.4 |
| B2 | 12.5 | **0.5** | 24 | – | – | – | 28.2 | -1.1 | 11.9 |
| B3* | 12.5 | **1.0** | 24 | – | – | – | 27.0 | 0.2 | 26.4 |
| B4 | 12.5 | **2.0** | 24 | – | – | – | 18.6 | -5.3 | 14.0 |
| B5 | 12.5 | **3.0** | 24 | – | – | – | 22.2 | -2.0 | 13.0 |
| B6 | 12.5 | **4.0** | 24 | – | – | – | 14.1 | -25.3 | -11.2 |
| B7† | 12.5 | **5.0** | 24 | – | – | – | – | – | – |
| B8 | 12.5 | **6.0** | 24 | – | – | – | 14.2 | 1.4 | 19.8 |
| Series C | | | | | | | | | |
| C1* | 12.5 | 1.0 | **24** | – | – | – | 27.0 | 0.2 | 26.4 |
| C2 | 12.5 | 1.0 | **30** | – | – | – | 17.7 | 2.0 | 21.7 |
| C3 | 12.5 | 1.0 | **36** | – | – | – | 14.7 | 1.5 | 22.3 |
| C4† | 12.5 | 1.0 | **42** | – | – | – | – | – | – |
| C5† | 12.5 | 1.0 | **48** | – | – | – | – | – | – |

**Table 2.** Parameter settings and results of ROPS runs in Series D. Bold numbers indicate those parameters which are varied within this series.

| Run | $t_{INI}$ | $\tau$ | $\overline{\Gamma_T}$ | $\overline{\Gamma_S}$ | $\overline{\Gamma_\sigma}$ | $\overline{\Gamma_\sigma}(D0)$ |
|---|---|---|---|---|---|---|
| | | [days] | [%] | [%] | [%] | [%] |
| D1 | **11 June** | **11.75** | 28.1 | 12.0 | 31.9 | 25.4 |
| D2 | **12 June** | **10.75** | 32.9 | 11.1 | 29.7 | 24.8 |
| D3 | **13 June** | **9.75** | 34.0 | 6.1 | 31.7 | 24.7 |
| D4 | **14 June** | **8.75** | 23.5 | 0.7 | 24.4 | 15.5 |
| D5 | **15 June** | **7.75** | 4.0 | -5.0 | 11.3 | 10.2 |
| D6 | **16 June** | **6.75** | 9.8 | -16.4 | 12.6 | 12.9 |
| D7 | **17 June** | **5.75** | 7.9 | -4.2 | 18.2 | 1.0 |
| D8 | **18 June** | **4.75** | 25.9 | -4.3 | 25.5 | 1.3 |
| D9 | **19 June** | **3.75** | 7.5 | -3.9 | 10.0 | -3.0 |
| D10 | **20 June** | **2.75** | -6.7 | -6.1 | 4.1 | 4.8 |
| D11 | **21 June** | **1.75** | -17.5 | -13.5 | -22.4 | -1.9 |
| D12 | **22 June** | **0.75** | 5.8 | -1.9 | 0.8 | 3.1 |

# Figures

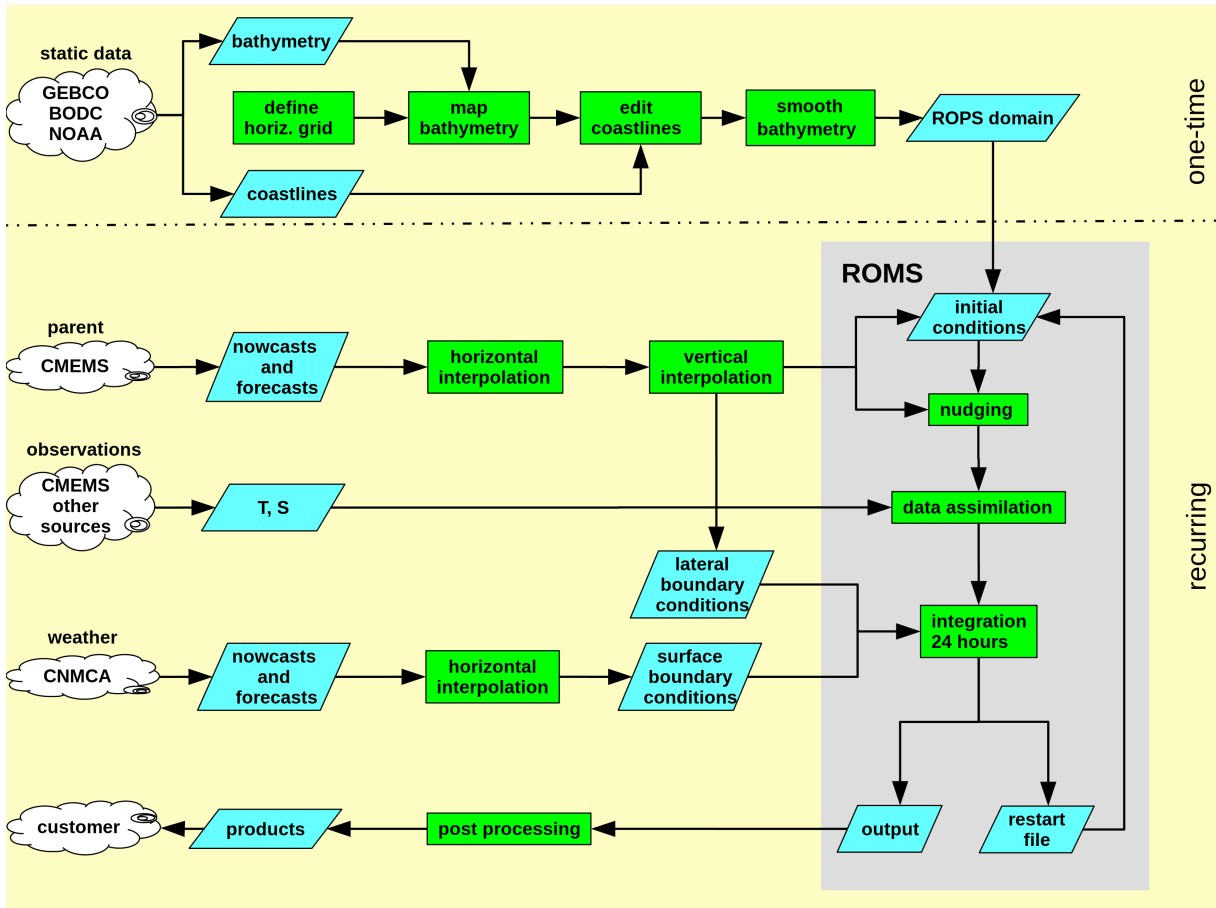

**Figure 1.** The ROPS concept: web resources are depicted by clouds, blue parallelograms represent data sets on the local host, processes are indicated by green rectangles. The processing of ROMS is accomplished within the grey box. $T$ and $S$ denote temperature and salinity, respectively.

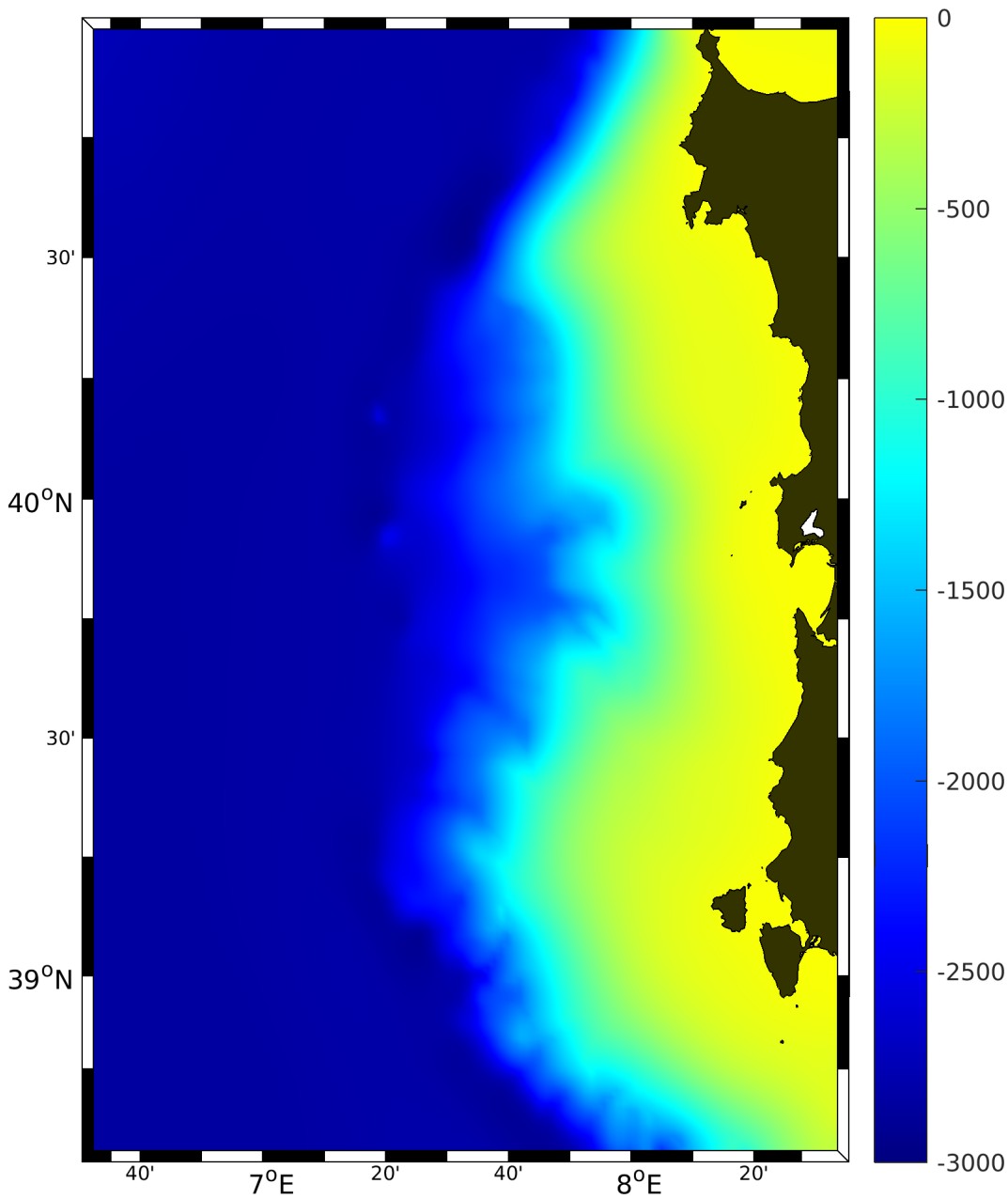

**Figure 2.** The ROPS domain; the colour code indicates the water depth [m] after smoothing.

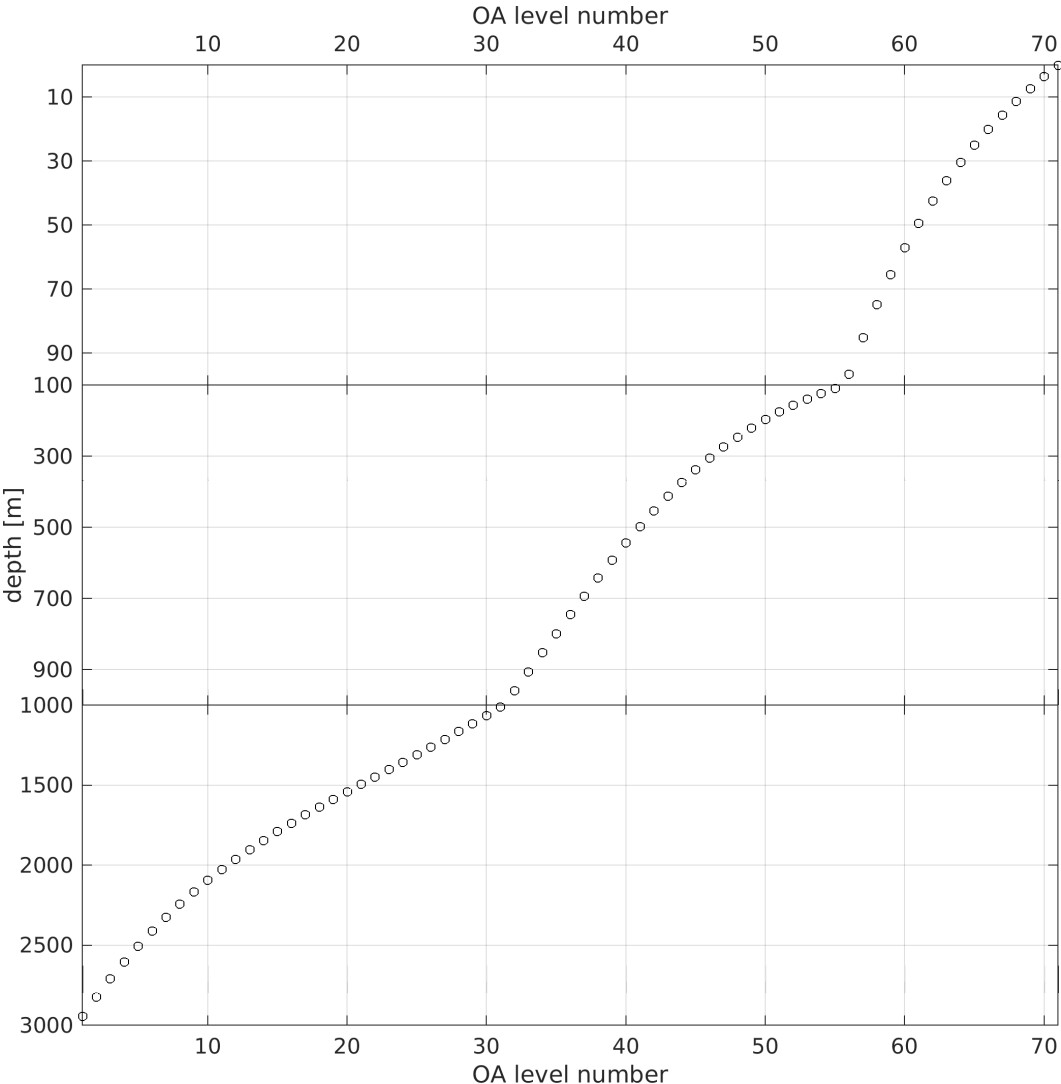

**Figure 3.** Depth of the vertical levels where the objective analysis (OA) is executed.

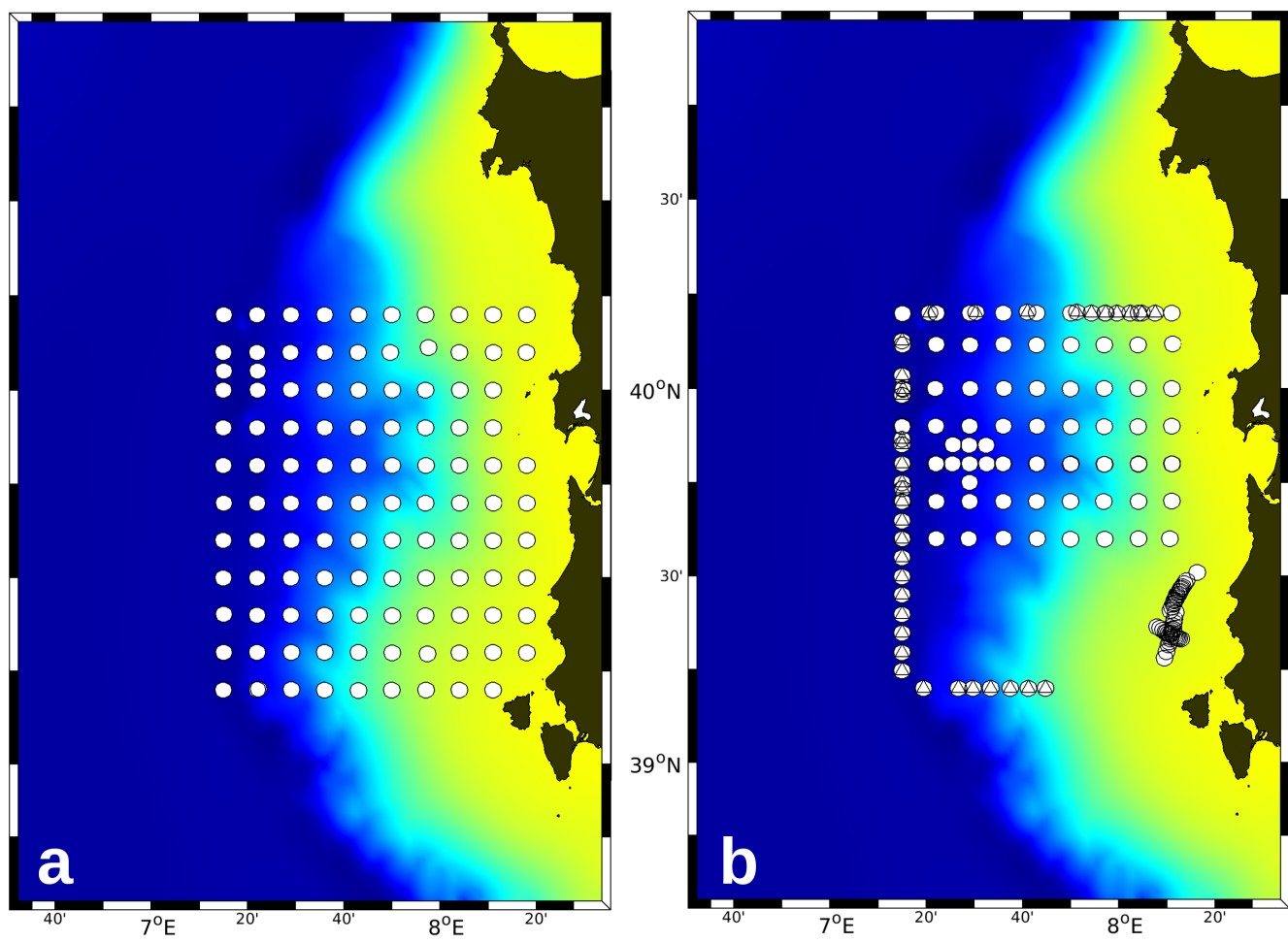

**Figure 4.** Positions of lowered CTD (circles) and underway CTD (triangles) casts collected during (a) Leg 1 (6–11 June) and (b) Leg 2 (12–20 June) of the REP14-MED experiment. The first casts were taken on 7 June. The colour code for the water depth is the same as in Fig. 2.

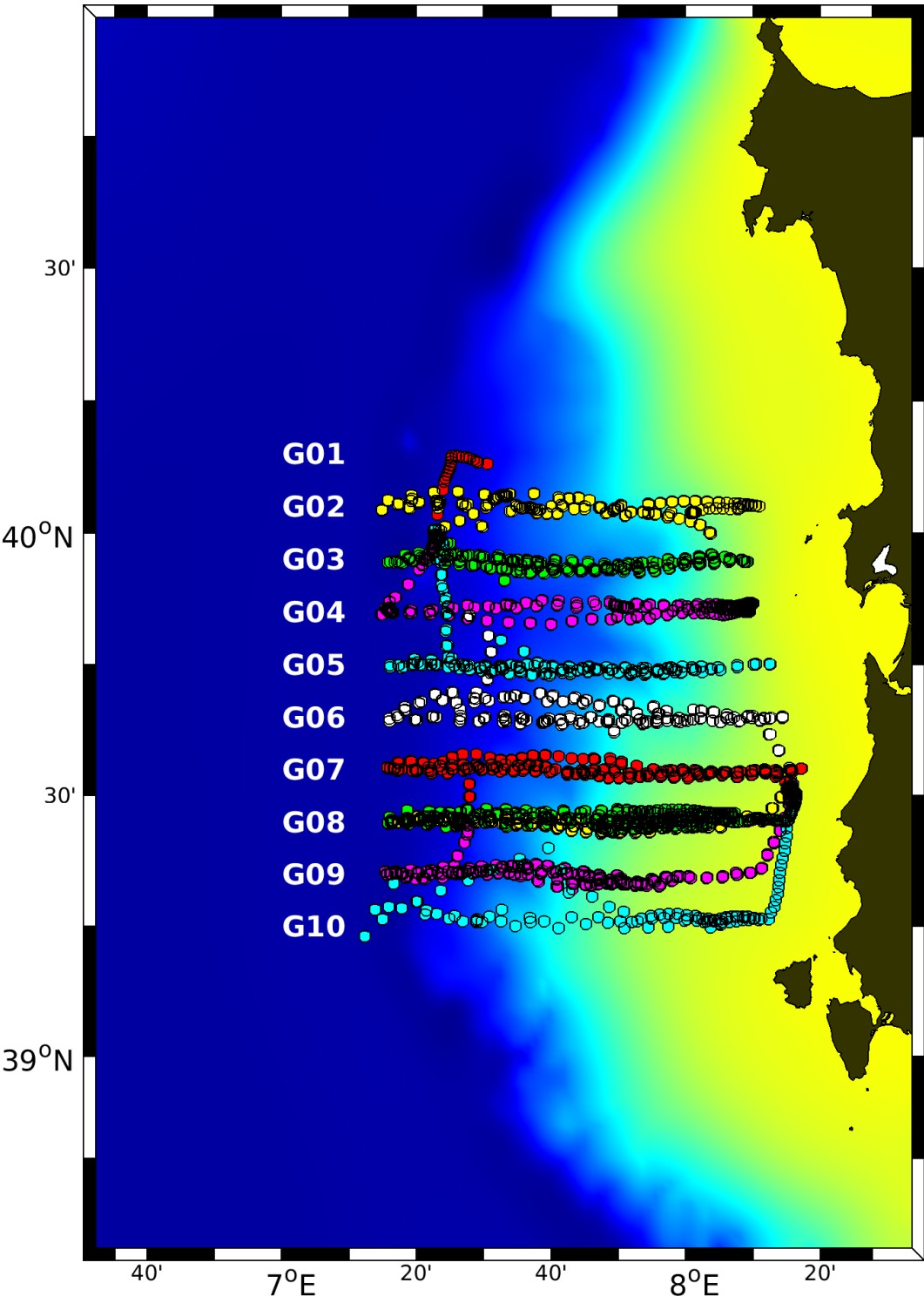

**Figure 5.** Surfacing positions of gliders, collected between 8 and 23 June. Each glider is marked by a different colour. The glider tracks are numbered G01–G10. G08 was occupied by 2 gliders. The colour code for the bathymetry is the same as in Fig. 2.

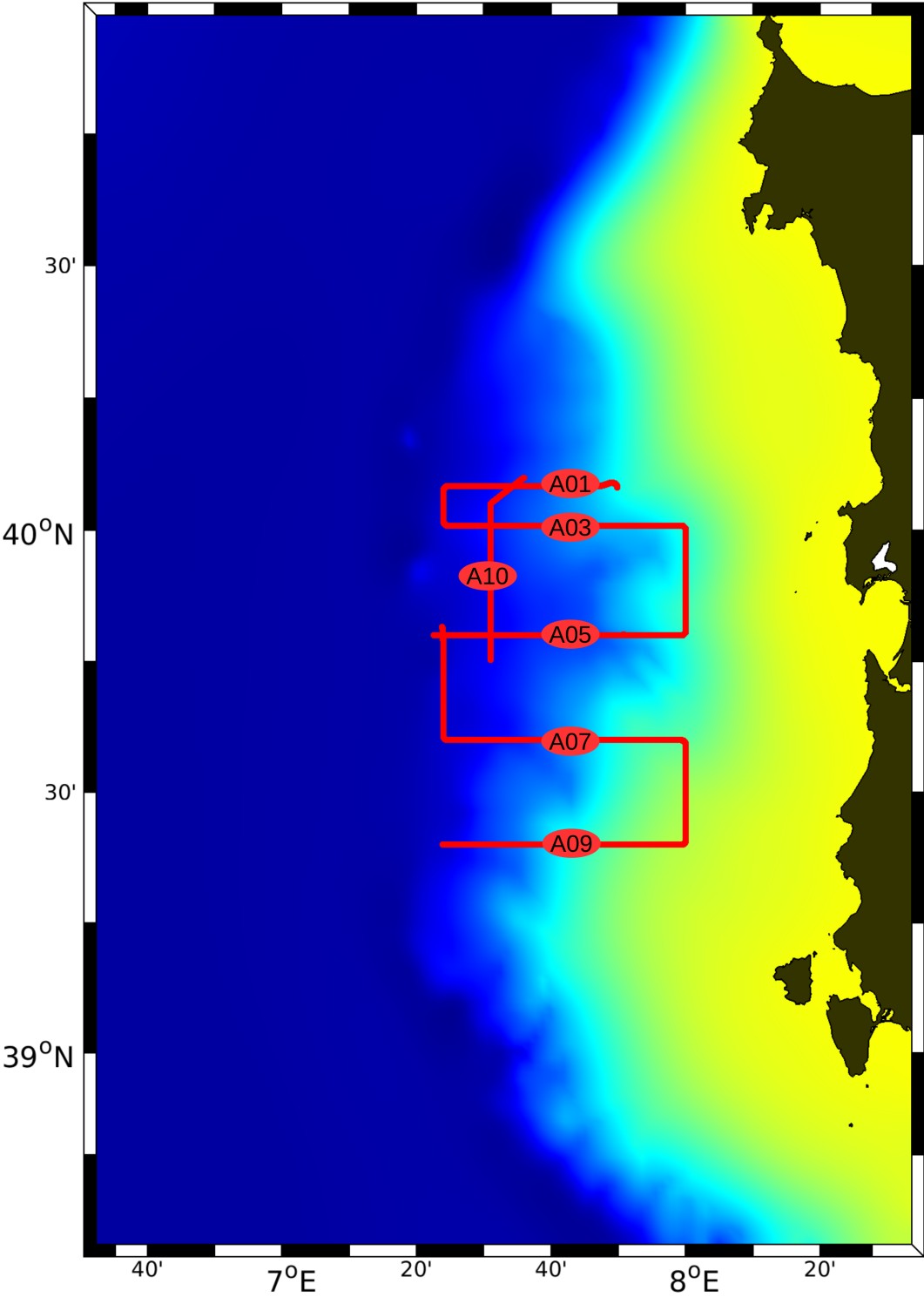

**Figure 6.** Tracks of the ScanFish tows (21– 23 June) of the REP14-MED experiment.

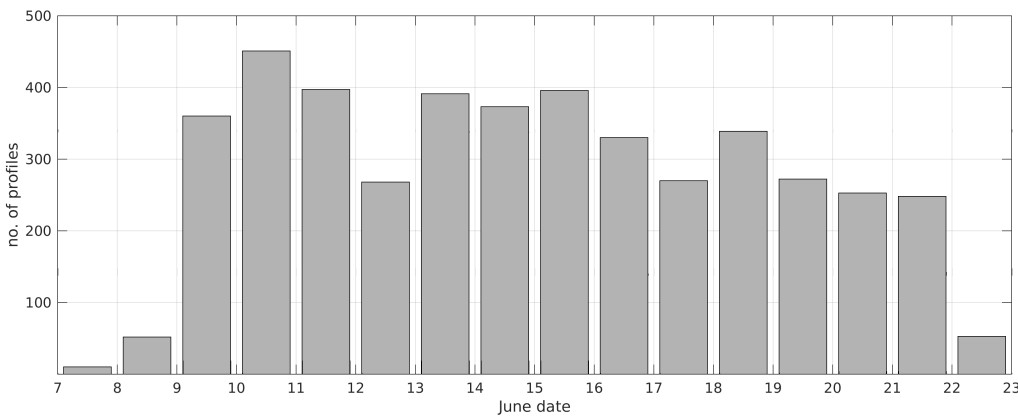

**Figure 7.** Number of profiles available for assimilation during the period 7–23 June. Profiles from shipborne CTD probes, underway CTD, and gliders are included. The dates on the abscissae indicate the start of each day at 00:00;

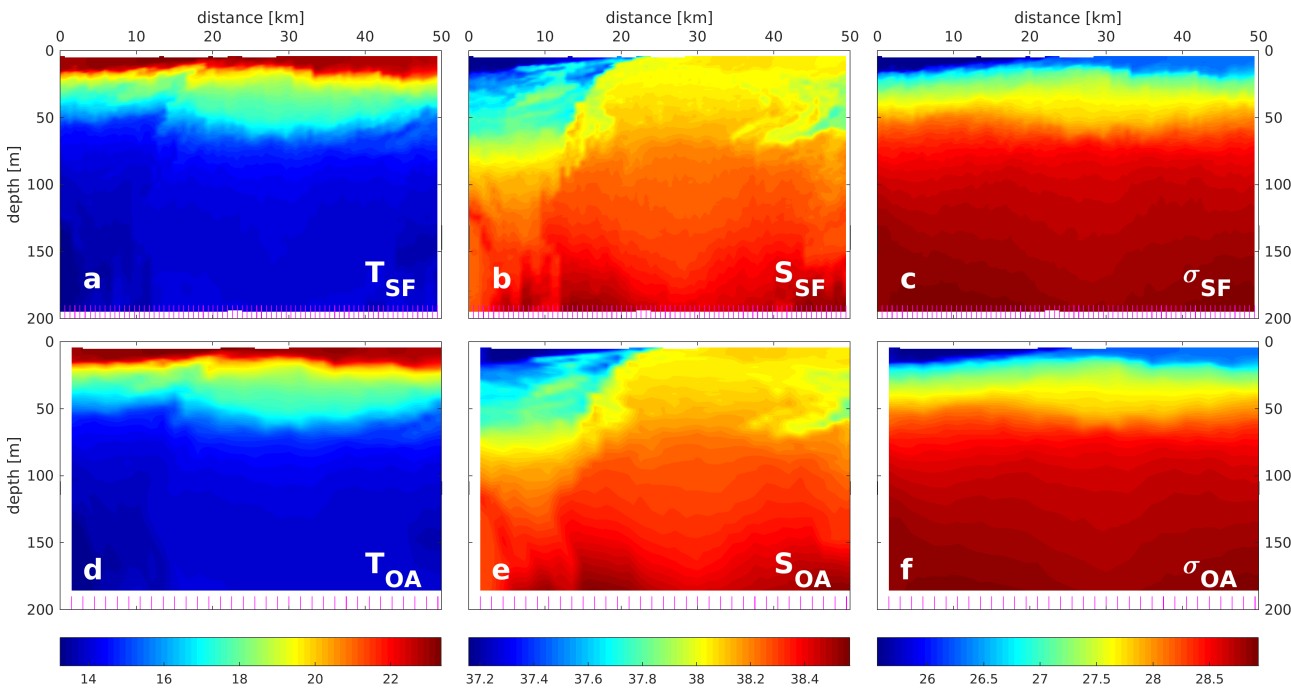

**Figure 8.** Observed (a) temperature [°C], (b) salinity, and (c) potential density [kg m$^{-3}$] along the southernmost zonal ScanFish (SF) section A09 (cf. Fig. 6). The positions of the ScanFish profiles are indicated by the magenta tick marks along the lower x-axis. Gridded (d) temperature, (e) salinity, and (f) potential density using objective analysis (OA). The tick marks indicate the OA grid. Only T and S underwent OA while potential density was computed from T, S, and depth.

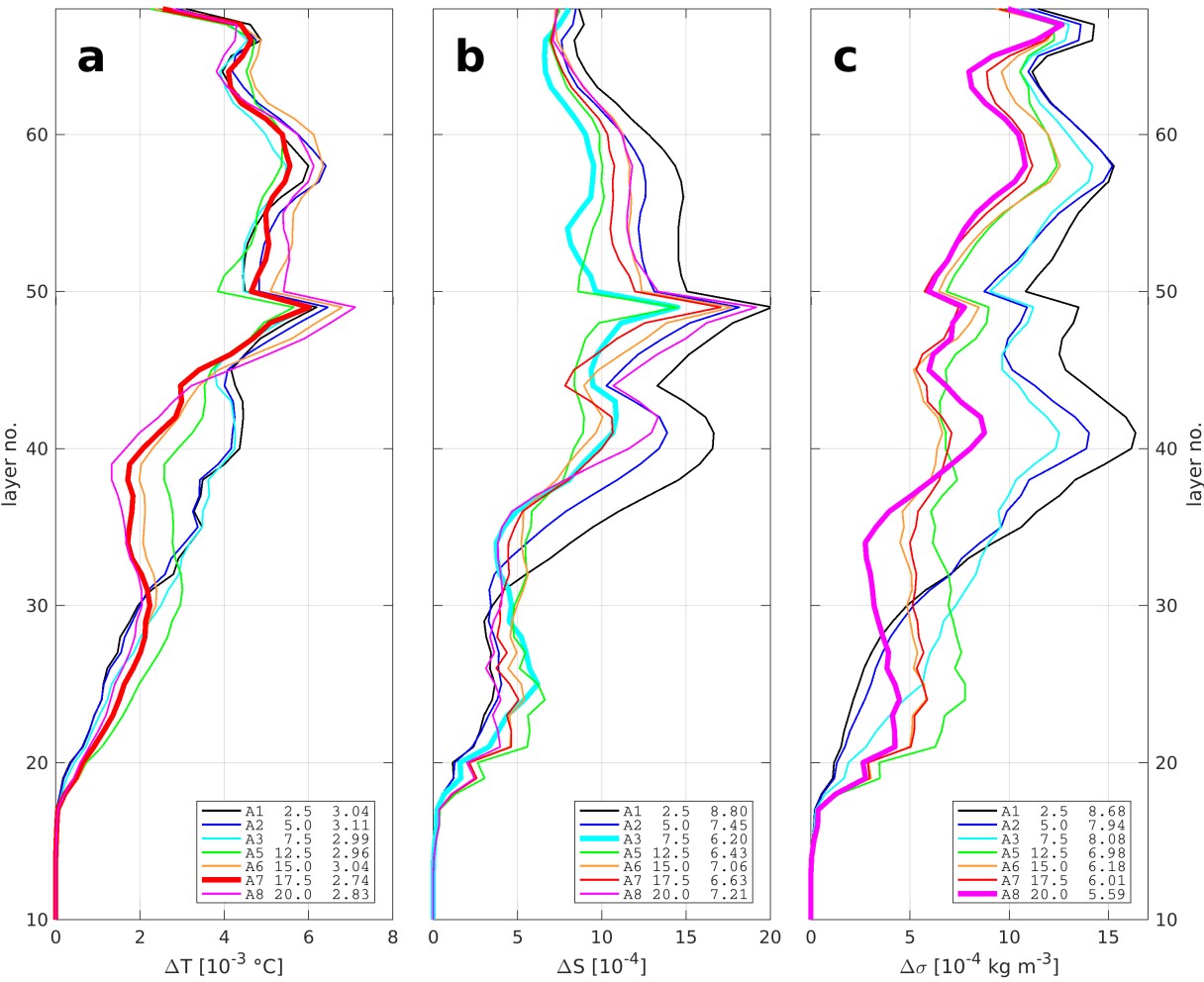

**Figure 9.** The vertical distributions of (a) $\Delta T$, (b) $\Delta S$, and (c) $\Delta \sigma$, for ROPS runs A1–A8. The first column in the legend boxes refers to the number of the ROPS run, the second column is the selected correlation scale $C$ [km], and in the third column is written the layer thickness-weighted mean $\overline{\Delta \Psi}$ where $\Psi$ stands for either tracer $T$, $S$, or $\sigma$. For better readability, $\Delta T$ was multiplied by $10^3$, and $\Delta S$, $\Delta \sigma$ by $10^4$. The bold graphs indicate the runs where $\overline{\Delta \Psi}$ were minimal.

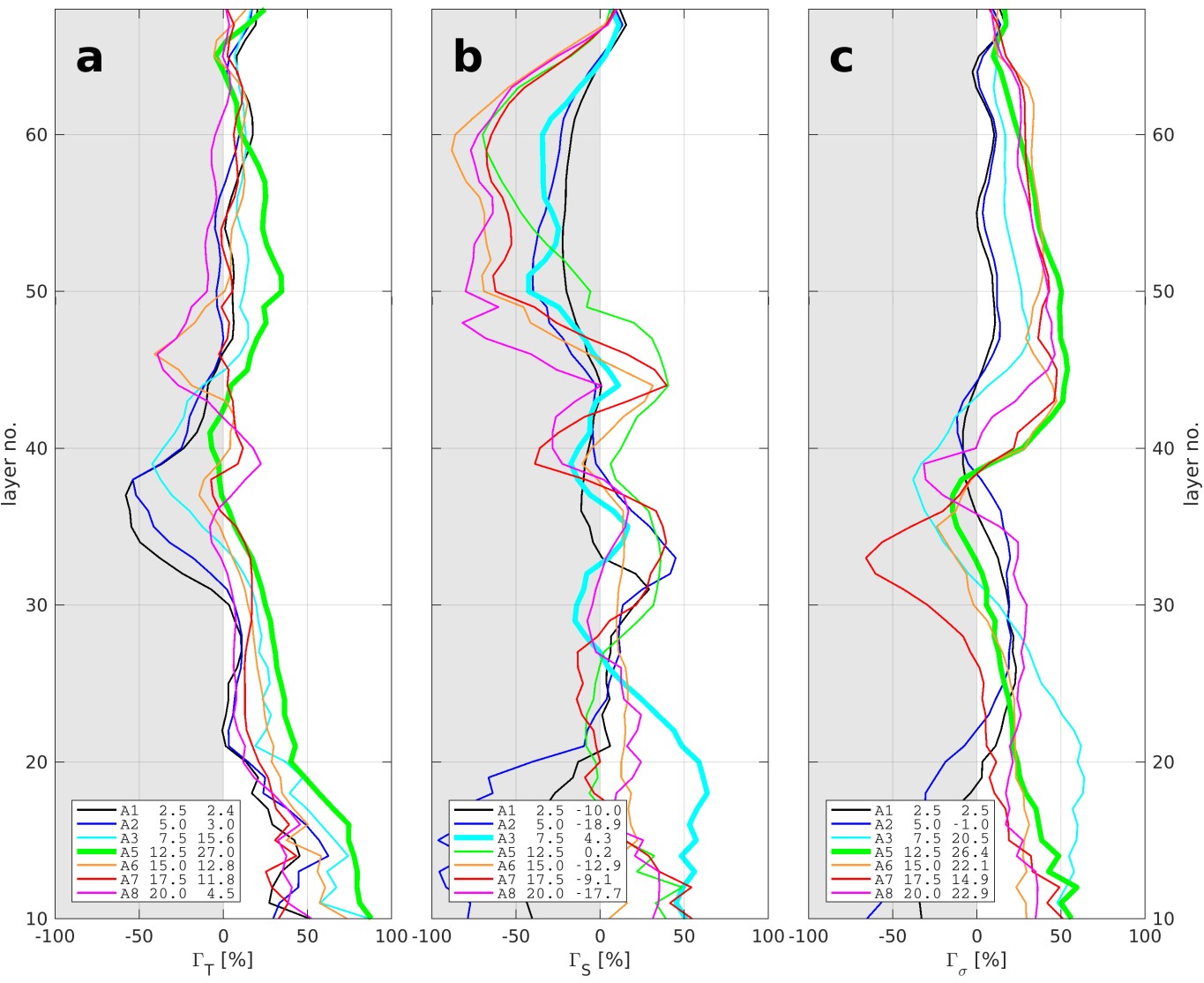

**Figure 10.** The vertical distributions of (a) $\Gamma_T$, (b) $\Gamma_S$, and (c) $\Gamma_\sigma$, for ROPS runs A1–A8. The first column in the legend boxes refers to the number of the ROPS run, the second column is the selected correlation scale $C$ [km], and in the third column is written the layer thickness-weighted mean $\overline{\Gamma_\Psi}$ where $\Psi$ stands for either tracer $T$, $S$, or $\sigma$. The bold graphs indicate the runs where $\overline{\Delta\Psi}$ were maximal.

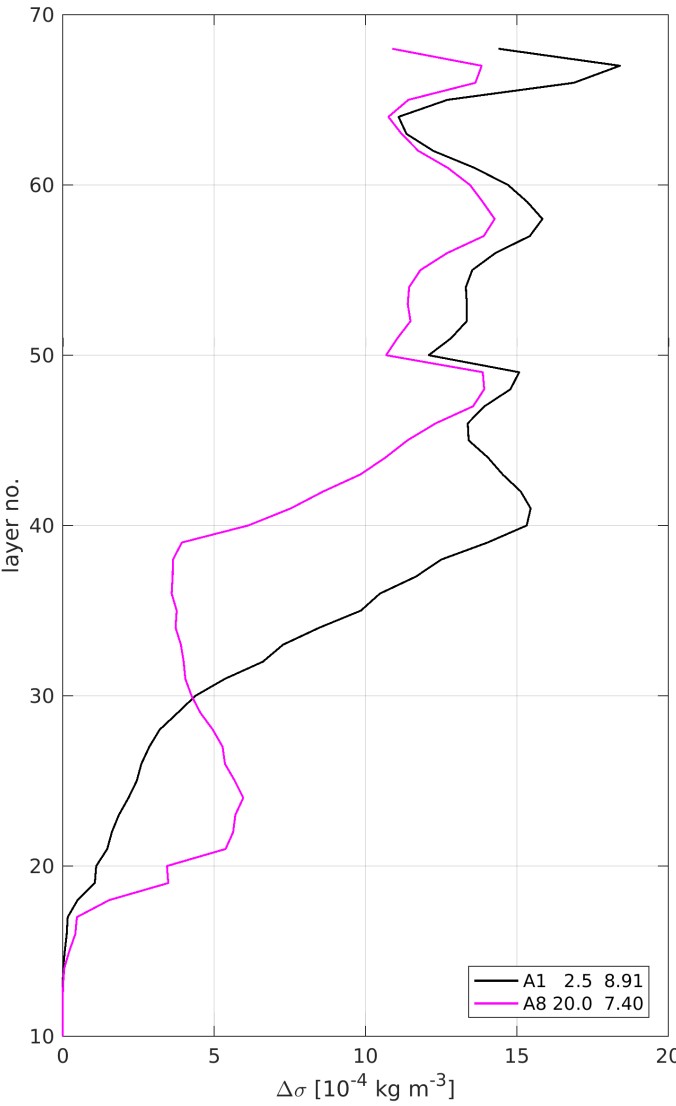

**Figure 11.** The vertical distributions of $\Delta\sigma(REF, OBS)$ for ROPS runs A1 and A8 at $t = t_{INI}$=18 June 00:00. The first column in the legend boxes refers to the number of the ROPS run, the second column is the selected correlation scale $C$ [km], and in the third column is written the layer thickness-weighted mean $\overline{\Delta\sigma}$. For better readability, $\Delta\sigma$ was multiplied by $10^4$.

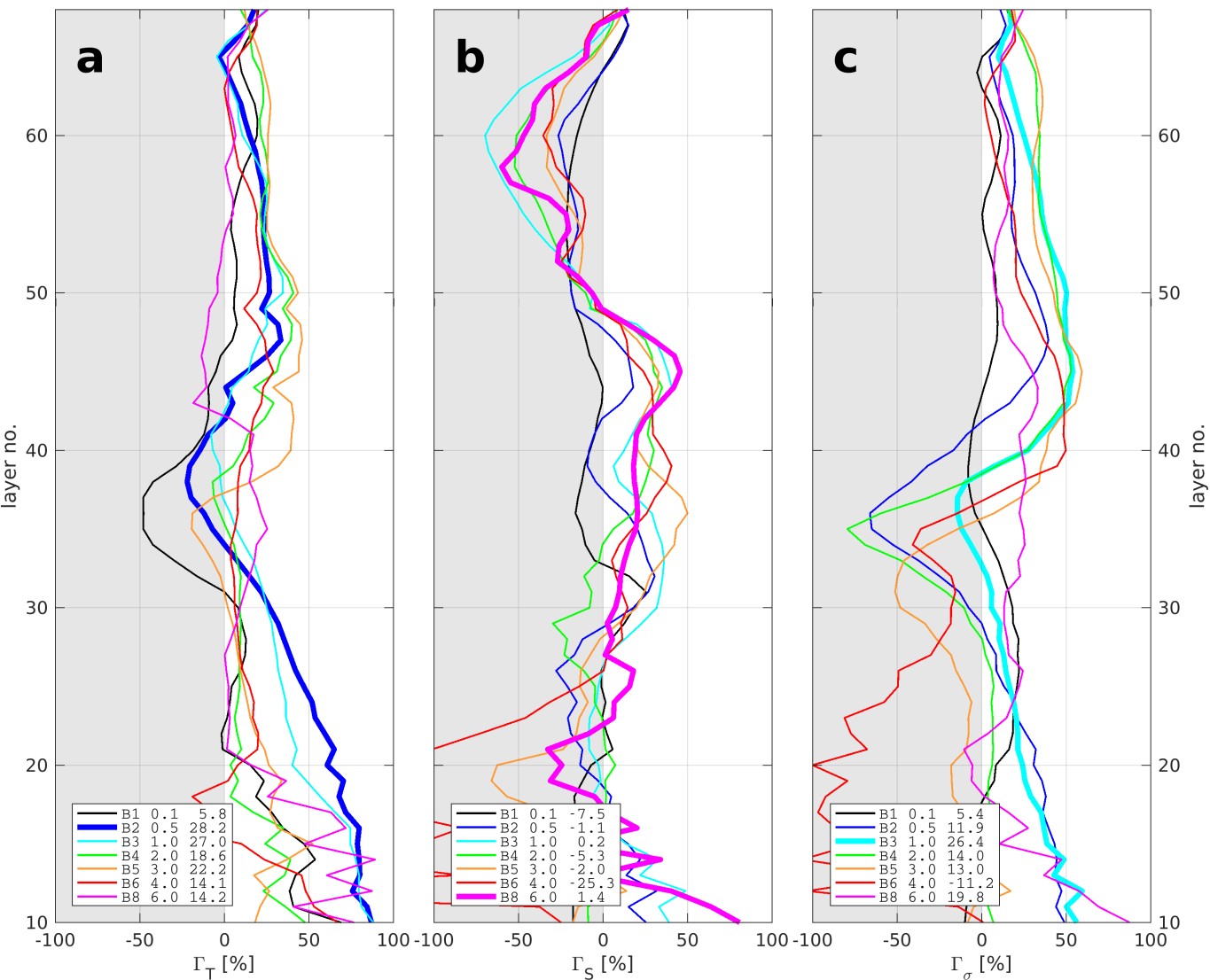

**Figure 12.** The vertical distributions of (a) $\Gamma_T$, (b) $\Gamma_S$, and (c) $\Gamma_\sigma$, for ROPS runs B1–B8. The first column in the legend boxes refers to the number of the ROPS run, the second column is the selected ratio $\delta\Psi_b/\delta\Psi_{obs}$ (for $\Psi \in T, S$), and in the third column is written the layer thickness-weighted mean $\overline{\Gamma_\Psi}$ where $\Psi$ stands for either tracer $T$, $S$, or $\sigma$. The bold graphs indicate the runs where $\overline{\Delta\Psi}$ were maximal.

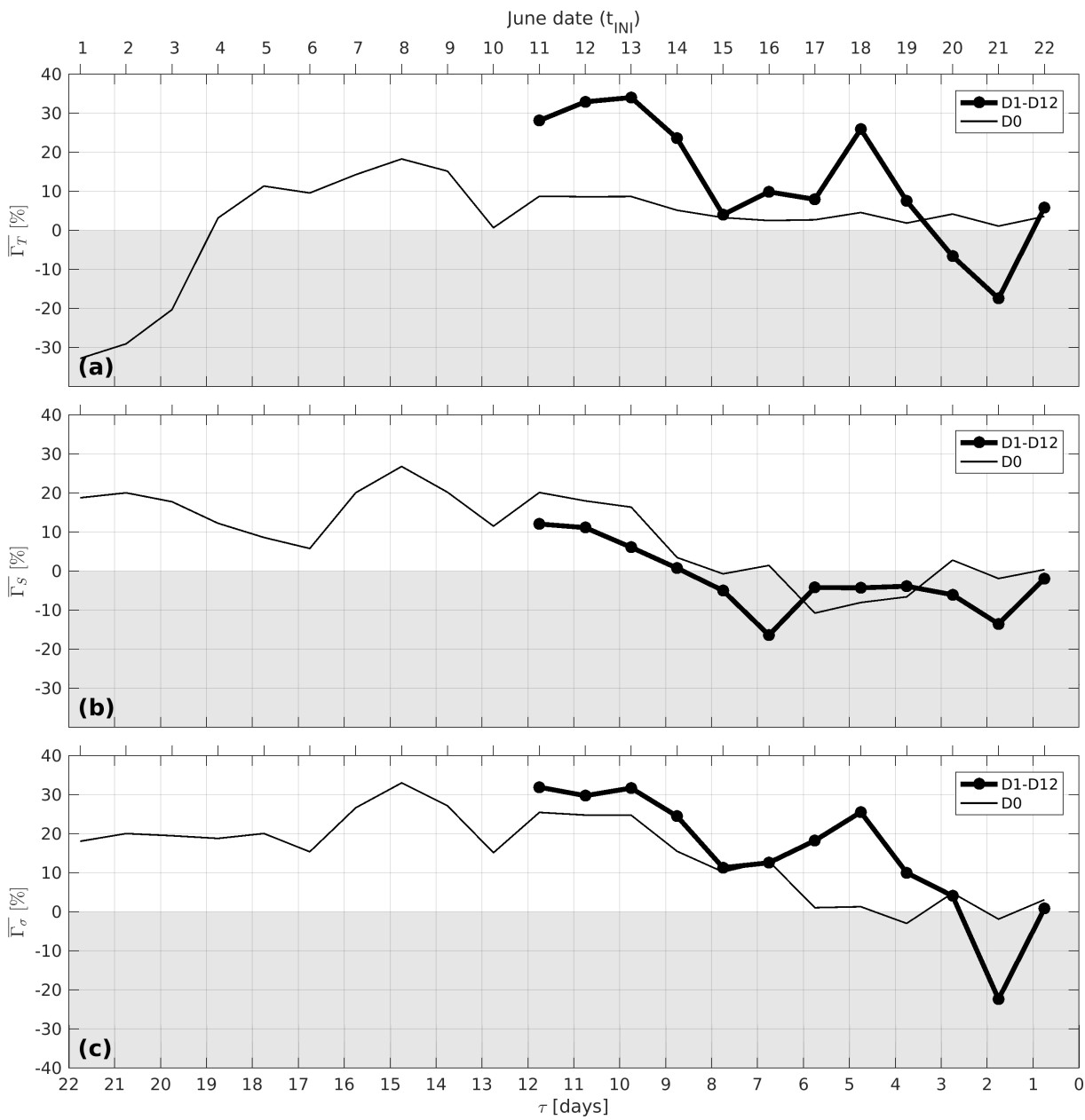

**Figure 13.** The skill scores (a) $\Gamma_T$, (b) $\Gamma_S$, and (c) $\Gamma_\sigma$ vs. the forecast range $\tau$ for ROPS runs D0 and D1–D12. The June dates on the top abscissae indicate the start of each day at 00:00; the dates are identical with the assimilation time $t_{INI}$. Note that the time axis at the top is offset by 6 hours with respect to the time axis at the bottom in order to synchronise $t_{INI}$ and $\tau$.

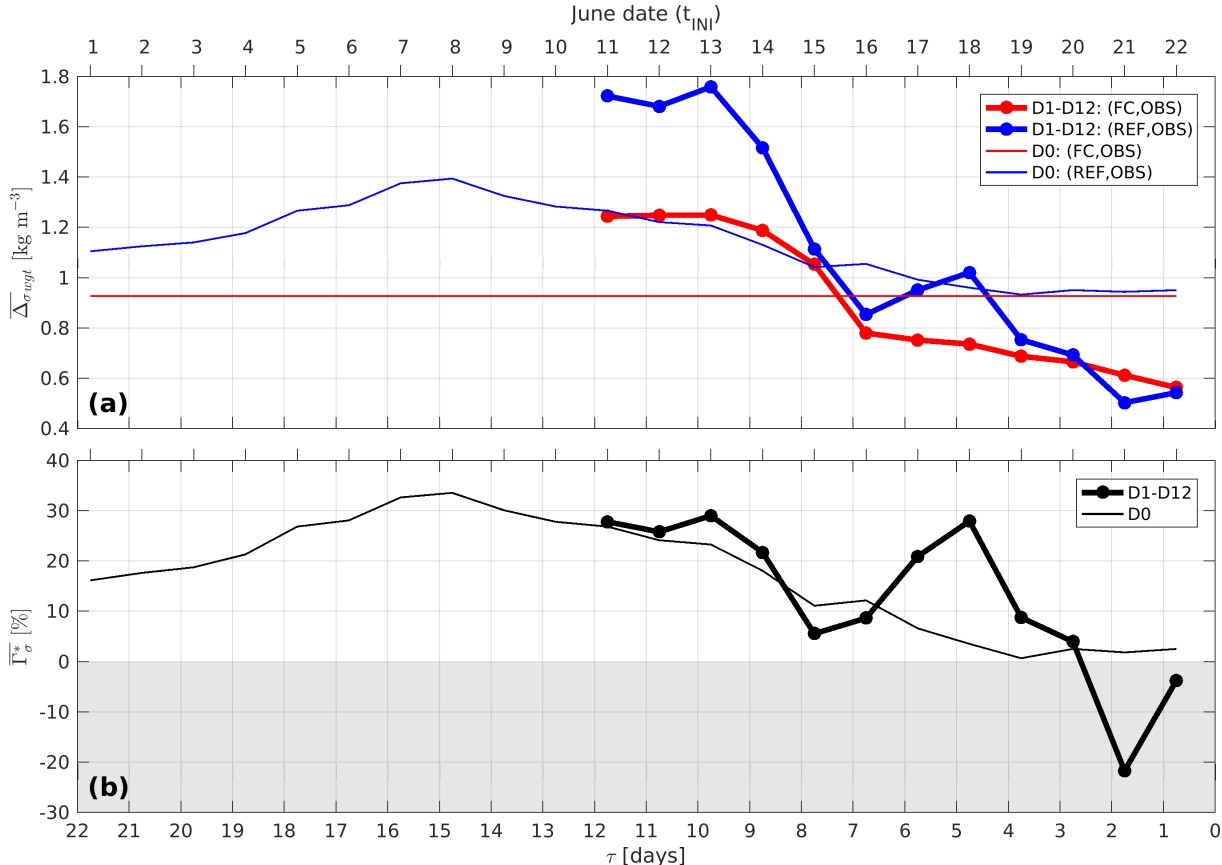

**Figure 14.** (a) $\overline{\Delta\sigma}_{wgt}(FC,OBS)$, $\overline{\Delta\sigma}_{wgt}(REF,OBS)$ and (b) $\overline{\Gamma^*_\Psi}$ (cf. eq. (6)) vs. the forecast range $\tau$ for ROPS runs D0 and D1–D12. The June dates on the top abscissae indicate the start of each day at 00:00; the dates are identical with the assimilation time $t_{INI}$. Note that the time axis at the top is offset by 6 hours with respect to the time axis at the bottom in order to synchronise $t_{INI}$ and $\tau$.