# Peer review of "Forecast skill score assessment of a relocatable ocean prediction system, using a simplified objective analysis method"

_Ocean Science, 2017_

## Referee Comment (RC1) · Anonymous Referee #1 · 24 Jul 2017

Title: Forecast skill score assessment of a relocatable ocean prediction system, using a simplified objective analysis method Author: Reiner Onken

GENERAL comments

The paper presents the Relocatable Ocean Prediciton System (ROPS) which is implemented in the Western Mediterranean Sea. ROPS is one-way nested in the MERCA-TOR global ocean model by downscaling. The model has assimilated, with a simplified objective analysis (OA) method, observational data collected in June 2014 in the mainframe of the REP14-MED experiment. A sensitivity study with respect to four OA parameters is then performed. The objectives exposed in the abstract are well achieved

and the work is well structured and organized. Minor revision are suggested to the author, as described below.

SPECIFC comments

• @page 1 rows 8:9 – In addition to (De Dominicis et al. 2014) and Rowley and Mask, 2014 I suggest to cite Trotta et al. 2016 as another recent work on Relocatable Ocean Prediciton System based on NEMO HD code i.e. Trotta et al (2016) A Structured and Unstructured Relocatable ocean model for Forecasting (SURF). Deep Sea Research Part II: Top. Stud. Oceanogr., 133, pp.54-75

• @page 3 sect.2.1 – Specify also which numerical schemes have been used for momentum advection and advection of tracers.

• @page 4 row 17 - The grid spacing ratio parent/child is set to 6.2. Why this value and how it is compared with other dynamical downscaling studies?

• @page 4 rows 28:33 - Are the interpolated fields adjusted in order to prevent violation of the continuity equation?

• @page 6 row 10: - How much spin-up period is needed for the child model?

• @page 6 row 13 - How are the eddy viscosity e diffusivity coefficient of the child related to the parent model? Do the child/parent models use the same lateral subgrid-scale mixing scheme?

Technical corrections.

• @page 1 row 8 – The citation De Dominicis, 2013 should be De Dominicis et al. 2014 (check references list).

• @page 5 row 8 – the sentence 'iss 1 hour' should be "is 1 hour"

• In the vertical distribution of the skill score (SS) (fig 9,11) increase the thickness of the vertical dot line (SS=0) in order to better distinguish the positive and negative value

of the SS or you can use a contrasting background colour as in fig.12.

The series of experiments are well done and I suggest the author to address the above points and re-submit the paper.
* * *

---

## Referee Comment (RC3) · Anonymous Referee #2 · 14 Sep 2017

The author presents a numerical study, based on a Relocatable Ocean Prediction System (ROPS) and in situ hydrological measurements west of Sardinia. The data assimilation method is Optimal Interpolation, the assimilated data are temperature (T) and salinity (S) The main goal is to assess the global forecast skill for T and S at time scales of several days. Another objective is to evaluate the sensibility of the forecast skill to parameters of the assimilation system.

I appreciate the very didactic way the assimilation platform is described which allows a clear understanding of most of the system components and of the implementation efficiency . The emphasis is put on pragmatic issues (relocatability, calculation possible

on a laptop, etc.) and this provides a clear and consistent conducting line through the paper. The experimental protocol for each series of tests is well explained and justified. The paper is well written.

My main concerns relate to the two following issues:

1/ the analysis of the results lacks from physical interpretation in terms of circulation processes. The latter would allow a better understanding of 'what the assimilation is effectively doing' and therefore of the results of the sensitivity tests. Even though the goal is to evaluate the performance of a relocatable system, I believe the evaluation process cannot be done without considering the specific dynamics of the study area.

2/ I do not understand how the vertical grid is handled. On which vertical levels is the OA performed: on the ROMS grid levels or on constant depth levels ? The verification method is based on spatial averages of the RMSE at different levels (eg figures 8 to 11): are these levels the ROMS levels (implying that the RMSE for different depths are spatially averaged) ? There is an absolute need to clarify these points; I believe a graph would greatly help.

The paper can be published provided that the two issues above are addressed. No extra calculation is required. I therefore recommend the author to address these issues (and see other remarks below) and then to resubmit the manuscript.

For issue 1, I suggest the following:

- make clear in your introduction what are the specifics space and time scales of variability that this study is targeting.

- add a short paragraph introducing the main circulation patterns of the study area over June 2014

- use this information to discuss or justify some choices or hypothesis: for instance synopticity is assumed for the ScanFish observations over 60 hours while it is found in series C results that data within a 42 and 48h window are too old or too far in the future

to be consistent with the model forecast at the central time.

- add a comment in the discussion on the fact that you do not distinguish the shelf and deep region, although these areas are characterized a priori by different scales of variability. If this is not the case in this specific Mediterranean area in June 2014, it needs to be stated. The differences in dynamical regimes are likely to explain some results on the behavior of the assimilation.

Other remarks:

- Series D: as the distribution of the assimilated data is not stationary in time, can this influence the results? For instance, leg 1 and leg 2 do not have exactly the same sampling pattern nor the same density of observations at the same location (if I understand well figure 3).

- Series D: the skill is relatively low for short forecast range with respect to longer forecast range in both assimilated and free runs: could this be due to errors at short time scales on the atmospheric forcing at the period of the verification ( around June 22)?

- As T and S are assimilated independently from each other and since the assimilation is performed independently at each level (as far as I understood) there is no constraint on the water masses. A T/S diagram, for the free run versus the obs and versus the assimilation run would allow to check that new unrealistic water masses are not created by the assimilation

- Section 5 (p14, l22-23) 'it is the massive amount of assimilation data which desequilibrates the terms of governing equations of ROMS ..'. The errors on observations are supposed uncorrelated: is this hypothesis valid with such a number of data ?

Minor revision points

About the choice for the correlation: Please indicate the correlation function for the spatial correlation.

Section 3: please indicate the max depths of the profiles from CTD, gliders and Scan-Fish measurements.

---

## Author Comment (AC2) · 20 Sep 2017

Interactive reply to Anonymous Referee #2 of manuscript OS-2017-35 "Forecast skill score assessment of a relocatable ocean prediction system, using a simplified objective analysis method" by Reiner Onken

The author presents a numerical study, based on a Relocatable Ocean Prediction System (ROPS) and in situ hydrological measurements west of Sardinia. The data assimilation method is Optimal Interpolation, the assimilated data are temperature (T) and salinity (S) The main goal is to assess the global forecast skill for T and S at time scales of several days. Another objective is to evaluate the sensibility of the forecast skill to parameters of the assimilation system.

I appreciate the very didactic way the assimilation platform is described which allows a clear understanding of most of the system components and of the implementation efficiency . The emphasis is put on pragmatic issues (relocatability, calculation possible on a laptop, etc.) and this provides a clear and consistent conducting line through the paper. The experimental protocol for each series of tests is well explained and justified. The paper is well written.

My main concerns relate to the two following issues:

1/ the analysis of the results lacks from physical interpretation in terms of circulation processes. The latter would allow a better understanding of 'what the assimilation is effectively doing' and therefore of the results of the sensitivity tests. Even though the goal is to evaluate the performance of a relocatable system, I believe the evaluation process cannot be done without considering the specific dynamics of the study area.

**2/ I do not understand how the vertical grid is handled. On which vertical levels is the OA performed: on the ROMS grid levels or on constant depth levels ?**

See page 5 lines 23–24: "The vertical levels are defined where the OA is performed; these levels are given by the depth of the s-coordinates at the maximum depth of the domain."

**Provisional Action**

- A new Fig. 3 will be added in the revised manuscript, see right.
- Reference to new Fig. 3 will be added in Section 2.6 after line 24

Fig. 3: Depth of the vertical levels where the objective analysis (OA) is executed.

The verification method is based on spatial averages of the RMSE at different levels (eg figures 8 to 11): are these levels the ROMS levels (implying that the RMSE for different depths are spatially averaged) ? There is an absolute need to clarify these points; I believe a graph would greatly help.

Yes – these are the ROMS levels. See manuscript

- Page 8 line 10: "Finally, the analysed fields were interpolated from the horizontal OA levels on the ROMS vertical grid."
- Page 9 line 16: "These quantities are plotted vs. the ROMS layer number"

**Provisional action**

- On page 9 around line 15, a hint will be added that  $\Delta T$ ,  $\Delta S$  and  $\Delta \sigma$  are evaluated on the ROMS vertical levels
- The graph: new Fig. 3 will be added (see above)

The paper can be published provided that the two issues above are addressed. No extra calculation is required. I therefore recommend the author to address these issues (and see other remarks below) and then to resubmit the manuscript.

For issue 1, I suggest the following:

- add a short paragraph introducing the main circulation patterns of the study area over June 2014

- make clear in your introduction what are the specifics space and time scales of variability that this study is targeting.

It is rather difficult to introduce the main circulation patterns in June 2014 without recalling the general circulation of the entire Western Mediterranean. Therefore, the classical circulation pattern and the water masses will be described first of all. Thereafter, the situation as found from the experiment in June 2014 is depicted and the targeted space and time scales are specified.

**Provisional action**

A new Fig. 1 and the following text will be added in the Introduction on page 3 after line 3: From a morphological point of view, the area of the ROPS model domain (Fig. 1) is characterised by a wide continental shelf area, the width of which varies between about 40 and 80 km. The shelf ends at water depths between 150 and 200 m, followed by the continental slope which features several canyons. The deep-sea area belongs to the Sardo-Balearic Basin and exhibits water depths of up to 2800 m. The general circulation of the area of interest was comprehensively described by Millot (1999) and this picture was still valid at the beginning of the experiment. Accordingly, the mean surface circulation is mainly related to the inflow of "new" Modified Atlantic Water (MAW) from the Strait of Gibraltar by means of anticyclonic eddies originating from the Algerian Current. Another branch of "old" MAW, which mixed with the underlying water masses on its large-scale cyclonic circulation through the Tyrrhenian, Ligurian, and Balearic Seas, comes probably from the west via the Balearic Current (García-Ladona et al., 1996). Just below the MAW, Winter Intermediate Water (WIW) follows the path of the MAW along its whole cyclonic path. WIW is formed in late winter in the northern and northwestern Provencal Basin and it is supposed that it also finds a direct way from the formation sites to the Sardo-Balearic Basin via mesoscale eddies. Levantine Intermediate Water (LIW) originates from the Eastern Mediterranean and the direct path to the ROPS domain is via the Sardinia Channel and then northward around the southern tip of Sardinia. Another LIW branch flows from the Strait of Sicily through the Tyrrhenian and Ligurian Seas into the Provencal Basin where it follows the cyclonic circulation paths of MAW and WIW.

LIW occupies the vertical range between the WIW and close to 1000-m. Below the LIW, Western Mediterranean Deep Water (WMDW) and Bottom Water (BW) are found. Finally, the North Balearic Front (Testor, 2003) represents the confluence zone between the waters coming from the south and the waters from the north; according to Fuda (2000) and Olita (2013), it is located between about 40° N and 41° N.

From careful analyses of the REP14-MED observational data set, it turned out that the distribution of the water masses and the circulation patterns resembled the classical picture described above, but there were also significant differences. According to the water mass analysis of Knoll et al. (2017), the temperature and salinity of MAW, LIW, and BW increased compared to the observations during the last decade. In addition, an anticyclonic WIW eddy with unusual low temperatures and salinities was identified which may confirm the existence of a direct route of WIW from its formation site to the observational site. By contrast to previous observations, LIW occupied the whole trial area and the predominant direction of the geostrophic flow was to the north with the largest transports in the deep water off the 1000-m depth contour; no LIW vein tied closely to the Sardinian coast was found south of 40° N. The MAW pattern was different: namely, the major northward transport occurred also to the west of the 1000-km depth contour in a broad 30 - 50-km wide band but in addition, there was a narrow vein of near-coastal northward currents, the width of which rarely exceeded 10 km. Southward transport with a zonal extend of 20 - 40 km prevailed between the 2 northward directed regimes. Both the meridional flow bands of MAW and LIW were connected by alternating 10 - 30-km wide zonal currents.

The observed geostrophic flow pattern suggests a mean transport to the north with superimposed mesoscale perturbations of 10 – 40 km in diameter. This defines another demand to ROPS to reproduce the horizontal variability of these scales, i.e. to resolve the Rossby radius. Concerning the temporal scales, repeated ADCP (Acoustic Doppler Current Profiler) sections indicate that noticeable changes of the flow field occur within 4 days (see Fig. 14 in Knoll et al., 2017). However, this time scale is stipulated by the minimum interval between the repeated ADCP surveys; in reality, shorter scales are likely. Hence, an additional demand is to resolve at least day-to-day changes.

---

## Author Comment (AC1)

**Interactive reply to Anonymous Referee #1 of manuscript OS-2017-35**
**"Forecast skill score assessment of a relocatable ocean prediction system, using a simplified objective analysis method" by Reiner Onken**

*The Prospective Action is subject to the comments of Referee #2 which are not yet available.*

**Page 1 rows 8–9: In addition to … I suggest to cite Trotta et al. 2016 …**
Good advice!
Prospective Action: The citation will be added in the revised version.

**Page 3 section 2.1 – Specify which numerical schemes have been used …**
For the horizontal advection of momentum, a third-order upstream bias advection scheme was used. A fourth-order, centered differences scheme was applied for the vertical advection (ROMS option UV_ADV).
For the horizontal and vertical advection of tracers, the ROMS default scheme, i.e. a 4-th order centred scheme with mono-harmonic mixing (option TS_DIF2) was applied.
Prospective Action: Specification of numerical schemes will be added in the revised version

**Page 4 row 17 – The grid spacing ratio parent/child is set to 6.2. Why this value and how it is compared with other dynamical downscaling studies.**
The setup of the ROMS domain in this study is identical to the setup in Onken (2017, Ocean Science, 13, 235–257) using a horizontal resolution of dx=1500 m. The selection of the horizontal resolution was driven by the following requirements:
- Resolution of mesoscale patterns; this demands proper resolution of the Rossby radius (~13 km in summer)
- Approximate resolution of the validation fields from the ScanFish survey (along-track distance between individual profiles 500–700 m)
- Make efficient use of the glider data (along-track resolution ~1000 m in deep water).

The only available parent models for nesting were MFS (dx ~7 km) and MERCATOR (dx~9.25 km). In Onken (2017) was shown that initialising ROMS from MERCATOR instead of MFS provided a better agreement between the modelled field and the observations. Therefore, MERCATOR was selected as parent model which yields a nesting ratio of 6.2. Namely, McWilliams (2016, Submesoscale currents in the ocean. Proc. R. Soc. A, 472, 20160117) states that "Experince shows … that the grid refinement factor should not be much larger than 3", but precursor tests of ROMS with dx=3000 m (nesting ratio 3.1) revealed no significant differences compared to the final version using dx=1500 m, except for that small mesoscale features were not at all resolved. This is in agreement with Pham et al. (2016, Optimizing dynamic downscaling in one-way nesting using a regional ocean model. Ocean Modelling, 106, 104–120) who showed that the magnitudes of errors were comparable, using nesting ratios of 3 or 6, respectively.
Other studies (e.g. Capet at al. 2008, J. Phys. Oceanogr., 38, 29–43; Gula et al., 2016, J. Phys. Oceanogr., 46, 305–325) used mostly nesting ratios ~3.
Prospective Action: A (short) discussion on these aspects will be included in the revised version.

**Page 4 row 28--33 – Are the interpolated fields adjusted in order to prevent violation of the continuity equation?**
ROMS offers lateral open boundary edge volume conservation switches to enforce *global* mass conservation of the child. These switches were set to ON along all open boundaries.
*Locally*, i.e. along the open boundaries, no extra adjustment of the interpolated fields is done. Plots of the vertical velocity revealed no abnormal behaviour of the vertical velocity along the open boundaries.
Prospective Action: none (?)

**Page 6 row 10 – How much spin-up period is needed for the child model?**
Please see the figure below which shows a time series of the kinetic energy (KE) of a ROMS run
without assimilation. From 1 to 10 June, KE increases continuously from 1.12 to 1.61, and
thereafter it fluctuates between < 1.6 and > 1.3. The e-folding time (63.2% level ⇔ KE=1.43) –
which may be considered as the spin-up period – is 7 days.  Hence, as the majority of observations
is assimilated after 8 June (compare Fig. 6 in the manuscript), the spin-up is almost completed at
that time.

Prospective Action: A remark will be added in the revised version.

[Figure]

**Page 6 row 13 – How are the eddy viscosity and diffusivity coefficient of the child related to
the parent model? Do the child/parent models use the same lateral subgrid-scale mixing
scheme?**
see table:

|  | MERCATOR | ROMS |
|---|---|---|
| scheme for mixing of tracers | Laplacian | Laplacian |
| diffusion coefficient | 80 m² s$^{-1}$ | 5 m² s$^{-1}$ |
| scheme for mixing of momentum | bi-harmonic | Laplacian |
| viscosity coefficient | $-10^{11}$ m⁴ s$^{-1}$ | 10 m² s$^{-1}$ |

What worries me is the negative viscosity in MERCATOR.

Prospective Action: ?

**Page 1 row 8 – Citation DeDominics**
Prospective Action: Will be corrected in the revised version

**Page 5 row 8 – 'iss 1 hour'**
Prospective Action: Will be corrected in the revised version

**Figs. 9, 11**
Prospective Action: Will be amended in the revised version

---

## Author Response (AR1)

[revised manuscript text omitted]

Author: Reiner Onken
* * *
*All page (P) and line (L) numbers refer to the revised manuscript except for if otherwise stated*
* * *
**Action taken on specific points raised by Anonymous Referee #1**

**GENERAL comments**
**The paper presents the Relocatable Ocean Prediciton System (ROPS) which is implemented in the Western Mediterranean Sea. ROPS is one-way nested in the MERCATOR global ocean model by downscaling. The model has assimilated, with a simplified objective analysis (OA) method, observational data collected in June 2014 in the mainframe of the REP14-MED experiment. A sensitivity study with respect to four OA parameters is then performed. The objectives exposed in the abstract are well achieved and the work is well structured and organized. Minor revision are suggested to the author, as described below.**
**SPECIFC comments**
**• @page 1 rows 8:9 – In addition to (De Dominicis et al. 2014) and Rowley and Mask, 2014 I suggest to cite Trotta et al. 2016 as another recent work on Relocatable Ocean Prediciton System based on NEMO HD code i.e. Trotta et al (2016) A Structured and Unstructured Relocatable ocean model for Forecasting (SURF). Deep Sea Research Part II: Top. Stud. Oceanogr., 133, pp.54-75**

Action
P2 L10 – 11: done.

**• @page 3 sect.2.1 – Specify also which numerical schemes have been used for momentum advection and advection of tracers.**

Action
P4 L9 – 11: the specification of the numerical schemes has been added .

**• @page 4 row 17 - The grid spacing ratio parent/child is set to 6.2. Why this value and how it is compared with other dynamical downscaling studies?**

Action
P5 L11 – 24: a discussion of this issue has been added

**• @page 4 rows 28:33 - Are the interpolated fields adjusted in order to prevent violation of the continuity equation?**

An explanation was given to the Referee in the OS Open Discussion.

Action
None (not required).

**• @page 6 row 10: - How much spin-up period is needed for the child model?**

Action
P7 L17 – 18: a discussion of this issue has been added.

**• @page 6 row 13 - How are the eddy viscosity e diffusivity coefficient of the child related to the parent model? Do the child/parent models use the same lateral subgrid-scale mixing scheme?**

An explanation was given to the Referee in the OS Open Discussion.

Action
None (not required).

**Technical corrections.**
**• @page 1 row 8 – The citation De Dominicis, 2013 should be De Dominicis et al. 2014 (check references list).**

Action
P2 L8: done.

**• @page 5 row 8 – the sentence 'iss 1 hour' should be "is 1 hour"**

Action
P6 L15: done.

**• In the vertical distribution of the skill score (SS) (fig 9,11) increase the thickness of the vertical dot line (SS=0) in order to better distinguish the positive and negative value of the SS or you can use a contrasting background colour as in fig.12.**

Action
Done. The contrasting background was used. See new Figures 10 and 12.

**The series of experiments are well done and I suggest the author to address the above points and re-submit the paper.**

Manuscript os-2017-35
Title:   Forecast skill score assessment of a relocatable ocean prediction system, using a simplified objective analysis method
Author: Reiner Onken
* * *
**Action taken on specific points raised by Anonymous Referee #2**

**The author presents a numerical study, based on a Relocatable Ocean Prediction System (ROPS) and in situ hydrological measurements west of Sardinia. The data assimilation method is Optimal Interpolation, the assimilated data are temperature (T) and salinity (S) The main goal is to assess the global forecast skill for T and S at time scales of several days. Another objective is to evaluate the sensibility of the forecast skill to parameters of the assimilation system.**
**I appreciate the very didactic way the assimilation platform is described which allows a clear understanding of most of the system components and of the implementation efficiency . The emphasis is put on pragmatic issues (relocatability, calculation possible on a laptop, etc.) and this provides a clear and consistent conducting line through the paper. The experimental protocol for each series of tests is well explained and justified. The paper is well written.**
**My main concerns relate to the two following issues:**
**1/ the analysis of the results lacks from physical interpretation in terms of circulation processes. The latter would allow a better understanding of 'what the assimilation is effectively doing' and therefore of the results of the sensitivity tests. Even though the goal is to evaluate the performance of a relocatable system, I believe the evaluation process cannot be done without considering the specific dynamics of the study area.**

**2/ I do not understand how the vertical grid is handled. On which vertical levels is the OA performed: on the ROMS grid levels or on constant depth levels ?**

See old ms P5 L23–24: "The vertical levels are defined where the OA is performed; these levels are given by the depth of the s-coordinates at the maximum depth of the domain."

Action
- New Fig. 3 has been  added
- P6 L30: reference to Fig. 3 has been added.

**The verification method is based on spatial averages of the RMSE at different levels (eg figures 8 to 11): are these levels the ROMS levels (implying that the RMSE for different depths are spatially averaged) ? There is an absolute need to clarify these points; I believe a graph would greatly help.**

Yes – these are the ROMS *layers* (not *levels*). See old manuscript
- P8 L10: "Finally, the analysed fields were interpolated from the horizontal OA levels on the ROMS vertical grid."
- P9 L16: "These quantities are plotted vs. the ROMS layer number"

Action
- P10 L33 – 34: a hint has been added that $\Delta T$, $\Delta S$ and $\Delta \sigma$ are evaluated in the ROMS vertical layers
- New Fig. 3 has been added (see above)

**The paper can be published provided that the two issues above are addressed. No extra calculation is required. I therefore recommend the author to address these issues (and see other remarks below) and then to resubmit the manuscript.**

**For issue 1, I suggest the following:**
**- add a short paragraph introducing the main circulation patterns of the study area over June 2014**
**- make clear in your introduction what are the specifics space and time scales of variability that this study is targeting.**

It is rather difficult to introduce the main circulation patterns in June 2014 without recalling the general circulation of the entire Western Mediterranean. Therefore, the classical circulation pattern and the water masses will be descibed first of all. Thereafter, the situation as found from the experiment in June 2014 is depicted and the targeted space and time scales are specified.

Action
- P3 L6 – 16: a paragraph describing the classical main circulation pattern has been added
- P3 L17 – 29: a paragraph describing the main circulation pattern as found in June 2014 has been added
- P3 L29 – 33: lines have been added on the space and time scales that this study is targeting

**- use this information to discuss or justify some choices or hypothesis: for instance synopticity is assumed for the ScanFish observations over 60 hours while it is found in series C results that data within a 42 and 48h window are too old or too far in the future to be consistent with the model forecast at the central time.**

Action
- P9 L17 – 21: a comment was added concerning the synopticity assumption for the ScanFish survey.
- P12 L17 – 18: a comment was added concerning the blow-up of runs C4 and C5

**- add a comment in the discussion on the fact that you do not distinguish the shelf and deep region, although these areas are characterized a priori by different scales of variability. If this is not the case in this specific Mediterranean area in June 2014, it needs to be stated. The differences in dynamical regimes are likely to explain some results on the behavior of the assimilation.**

In Section 4.2 (old ms P8 L31 – 34) is was stated
*"C was selected isotropic because a preliminary processing of data from shipborne Acoustic Doppler Current Profilers had revealed that the major part of the model domain was characterised by an eddy field with alternating currents; only along the west coast of Sardinia, predominantly meridional currents were prevailing in a ≈10-km wide stripe."*

This implies already that the different scales of spatial variability were recognised and – using a constant and isotropic correlation scale – that I did not take account of the different flow regimes. I have no idea how to discuss this in terms of "explaining some results on the behavior of the assimilation".

Action

None.

**Other remarks:**
**- Series D: as the distribution of the assimilated data is not stationary in time, can this influence the results? For instance, leg 1 and leg 2 do not have exactly the same sampling pattern nor the same density of observations at the same location (if I understand well figure 3).**

In principle you are right but you have to take into account that (old) Fig. 3 shows only the casts from shipborne CTDs and underway CTDs while the vast majority of CTD profiles originates from the gliders. In total from 8 to 23 June, there were
  • 312 profiles from lowered CTD (lCTD) and underway CTD (uCTD),
  • 5731 profiles from gliders.
Hence, the contribution of lCTD and uCTD profiles is only 5.2% of the total numbers of profiles and their non-stationarity in space and time is expected to have a negligible influence on the results. By contrast, the glider casts are more or less stationary in time (~ 1 yo/hour, dependent on the scheduled maximum diving depth) and space (meridional distance ~ 10 km, zonal distance ~ O(1) km depending on diving depth and water depth).

Action
None, because it was already stated in the old ms P12 L6 – 7:
*"As this feature was observed both for the assimilation runs D1–D12 and for the free run D0, it can be excluded that it was somehow caused by the assimilation of observational data."*

Including the above said statistics of profiles would make the discussion meaningless at this point.

**- Series D: the skill is relatively low for short forecast range with respect to longer forecast range in both assimilated and free runs: could this be due to errors at short time scales on the atmospheric forcing at the period of the verification ( around June 22)?**

For the atmospheric forcing, 3 different sources were available:
  • COSMO-ME (that was used)
  • COSMO-IT
  • observations from a meteorological buoy named M1 (point source)
In another paper (Onken, 2017a), the performance of all 3 forcings was evaluated and – surprisingly, the M1 observations melded with COSMO-ME did best, but the second-best performance was obtained from COSMO-ME. In the Fig. Fig_Forcing below are shown the 3 forcings and it can be seen that there are differences between COSMO-ME and COSMO-IT, especially for U and V after 21 June. Unfortunately, the M1 buoy was recovered on 20 June and there is no chance to asses which of the COSMO models did best after 20 June.

Yes, you are right – the low skill of the short forecast ranges could probably be due to errors of the forcing fields. However, as a similar beviour was also found by Ryan et al. (2015) and Tonani et al. (2009), I am quite confident that the low skills are not due to errors of the atmospheric forcing. Please see the discussion of this issue in the old ms P14, first paragraph.

Action
None.

[Figure]

*Fig_Forcing: Components of atmospheric forcing obtained from observations at the mooring M1, COSMO-ME and COSMO-IT.*

**- As T and S are assimilated independently from each other and since the assimilation is performed independently at each level (as far as I understood) there is no constraint on the water masses. A T/S diagram, for the free run versus the obs and versus the assimilation run would allow to check that new unrealistic water masses are not created by the assimilation**

In Fig_TS(a),(b) are shown T/S diagrams from the unforced run D0 and the assimilation run D8 were the last assimilation took place on 18 June. Both plots show the situation on 22 June 00:00h. Note that (a) and (b) exhibit the same features but the plot layers were switched. It becomes evident that *per se* the assimilation did not create any new water masses, but it has generally increased the salinity in almost the entire depth range. For comparison, (c) shows a T/S diagram from all CTD and glider data (from Knoll et al. 2017). The strange features at S < 37.5 originate from an Algerian eddy in MERCATOR which is not reflected by any observations (see also Juza et al. 2015 which investigates misfits of MERCATOR and MFS with observations). Moreover, as that eddy is outside the observational domain in the very south, the normalized mapping error $\varepsilon_\psi$ is close to one and ROMS has no chance to remove that error. One sees that the assimilation tried to find a compromise between the background field from MERCATOR and the observations, but it is not able to remove obvious errors in the background field.

P14 L35 – P15 L1: a remark *"it was verified that the data assimilation did not create any unrealistic water masses in those regions where nearby observations were available"* was added

[Figure]

*Fig_TS: (a)(b) T/S-diagrams on 22 June 2014 from run D0 (black dots) and D8 (red). (c) TS-diagram from all observations 7 – 23 June*

**- Section 5 (p14, l22-23) 'it is the massive amount of assimilation data which desequilibrates the terms of governing equations of ROMS ..'. The errors on observations are supposed uncorrelated: is this hypothesis valid with such a number of data ?**

At 00:00 UTC on 20, 21, and 22 June, about 280, 250, and 150 profiles, respectively, are assimilated (see old ms Fig. 6). The vast majority of these data originates from 10 gliders (one of the initial 11 gliders died already on 10 June), while just 26 profiles were from shipborne CTD casts during this period of time. Hence, if we ignore the contributions from the shipborne casts, we are left with 10 independent instruments. Moreover, as the remaining glider fleet consisted of 2 different brands (7 Slocums, 3 Seagliders) from 3 different institutions,  the assumption that the observational errors are uncorrelated is justified. Is that what you mean?

Action
None.

**Minor revision points**
**About the choice for the correlation: Please indicate the correlation function for the spatial correlation.**

A Gaussian function is used for the spatial correlation.

Action
P6 L22: the desired information was added.

**Section 3: please indicate the max depths of the profiles from CTD, gliders and Scan-Fish measurements.**

Action
- P8 L4 – 6: here, the desired information for the CTDs is provided.

- P8 L10 – 11: here, the desired information for the gliders is provided.
- P8 L12 – 13: here, the desired information for the ScanFish is provided.

---

## Author Response (AR2)

[revised manuscript text omitted]